# Interferon signalling and non-canonical inflammasome activation promote host protection against multidrug-resistant *Acinetobacter baumannii*

Fei-Ju Li, Lora Starrs, Anukriti Mathur [ID], Daniel Enosi Tuipulotu [ID], Si Ming Man [ID] & Gaetan Burgio [ID] ✉

Multidrug-resistant (MDR) *Acinetobacter baumannii* are of major concern worldwide due to their resistance to last resort carbapenem and polymyxin antibiotics. To develop an effective treatment strategy, it is critical to better understand how an *A. baumannii* MDR bacterium interacts with its mammalian host. Pattern-recognition receptors sense microbes, and activate the inflammasome pathway, leading to pro-inflammatory cytokine production and programmed cell death. Here, we examined the effects of a systemic MDR *A. baumannii* infection and found that MDR *A. baumannii* activate the NLRP3 inflammasome complex predominantly via the non-canonical caspase-11-dependent pathway. We show that caspase-1 and caspase-11-deficient mice are protected from a virulent MDR *A. baumannii* strain by maintaining a balance between protective and deleterious inflammation. Caspase-11-deficient mice also compromise between effector cell recruitment, phagocytosis, and programmed cell death in the lung during infection. Importantly, we found that cytosolic immunity - mediated by guanylate-binding protein 1 (GBP1) and type I interferon signalling - orchestrates caspase-11-dependent inflammasome activation. Together, our results suggest that non-canonical inflammasome activation via the (Interferon) IFN pathway plays a critical role in the host response against MDR *A. baumannii* infection.

*Acinetobacter baumannii* is a Gram-negative bacterium that has emerged as one of the most prevalent causative agents of nosocomial infections around the world[1], frequently leading to urinary tract infections, intensive care unit (ICU)-acquired pneumonia and septicemia[2,3]. In the USA alone, ICU-acquired *A. baumannii* pneumonia presents a considerable disease burden, being encountered in 5–10% of patients receiving mechanical ventilation[4], and resulting in a high fatality rate due to septicemia[5]. It is classified by the World Health Organization (WHO) as a member of the ESKAPE pathogens (*Enterococcus faecium, Staphylococcus aureus, Klebsiella pneumoniae, Acinetobacter baumannii, Pseudomonas aeruginosa* and *Enterobacter spp.*). Due to carbapenem and colistin antibiotic resistance, *A. baumannii* is listed amongst the strains that are critical for new therapeutic strategies by WHO[1]. Unfortunately, beyond combination therapies, there are currently no efficacious treatments against multidrug resistant (MDR) and extensively drug resistant (XDR) *A. baumannii*[6]. Targeting the host instead of the pathogen, solely or as a combination therapy, could potentially lead to novel avenues in

overcoming – and potentially further circumventing - MDR/XDR resistance. To identify potential host targets, a comprehensive characterisation of the host innate response to the MDR/XDR *A. baumannii* bacteria is required.

Once *A. baumannii* invades the host, stimuli such as pathogen-associated molecular patterns (PAMPs), dead cells, or irritants (danger-associated molecular patterns, DAMPs) are detected and the host mounts a protective inflammatory response via inflammasomes[7]. PAMPs and DAMPs are sensed by cytosolic inflammasome sensors such as Absent in Melanoma 2 (AIM2) or NOD-like receptors (NLRs such as NLRC4 and NLRP3[8]). Upon activation, these inflammasome sensors recruit the inflammasome adaptor protein apoptosis-associated speck-like protein containing a caspase activation and recruitment domain (ASC, also known as PYCARD)[9]. The resultant inflammasome complex activates caspase-1, which is required to induce cleavage of the pro-inflammatory cytokines pro-interleukin-1β (pro-IL-1β) and pro-interleukin-18 (pro-IL-18), as well as

Division of Immunology and Infectious Disease, The John Curtin School of Medical Research, the Australian National University, Canberra, Australian Capital Territory, Australia. ✉e-mail: Gaetan.burgio@anu.edu.au

the pro-pyroptotic factor, Gasdermin D (GSDMD)[10,11], which along with NINJ1[12] drives an inflammatory programmed cell death known as pyroptosis[13,14]. Previous work reported that the NLRP3 inflammasome via the 'canonical' caspase-1 pathway contributes to host defence against *A. baumannii* isolates in an intranasal infection model[15]. However, NLRP3 knockout mice were only partially protected against *A. baumannii* bacteria strains, suggesting the existence of additional protective mechanisms against the bacteria.

A 'non-canonical' NLRP3 inflammasome activation pathway has been reported and is dependent on caspase-11 (mice)[11] and caspase-4 and -5 (humans)[16]. This pathway is essential to defend against pathogens via interferon (IFN) signalling[17]. IFNs activate multiple host cell death pathways (pyroptosis and necroptosis)[18] suggesting a protective role for interferon inducible molecules against MDR *A. baumannii* infection. Recently, cytosolic immunity mediated through IFN inducible molecules such as the GTPase guanylate-binding-proteins (GBPs)[19] have been reported as an important host innate defence against bacteria. Of particular interest, guanylate binding protein 1 (GBP1) binds to the bacterial lipopolysaccharide (LPS), mediates the assembly of other GBPs and the recruitment and activation of caspase-4[20]. Previous works have demonstrated the protective role of GBPs against intracellular bacteria, extracellular bacteria[21] and parasites[22]. However, the mechanisms of non-canonical inflammasome activation via type I IFNs, how they mediate pyroptosis, and the role of GBP1-mediated caspase activation in response to *A. baumannii* is unknown. Characterisation and better knowledge of such mechanisms would drive future interventions against severe MDR *A. baumannii* infections targeting the inflammasome or type I IFN pathways.

Here we performed a comprehensive characterisation of the non-canonical inflammasome pathway in an acute sepsis model in response to a virulent *A. baumannii* MDR strain (ATCC BAA-1605) resistant to carbapenem and a lipooligosaccharide (LOS)-deficient strain resistant to polymyxin on the commonly used ATCC 19606 background strain[23,24]. We firstly found both canonical and non-canonical NLRP3 pathways were activated in response to a systemic and severe sepsis. We discovered that caspase-1 or caspase-11 single-deficiency conferred a protective effect against infection by promoting protective inflammation. Intriguingly, we found that the host utilises type I IFN signalling to mediate caspase-11 non-canonical NLRP3 inflammasome activation. Finally, we found that GBP1 promoted host resistance against *A. baumannii* strains, via inflammasome activation rather than direct bacteria killing. Together these findings underscore the requirement of caspase-11 and the type I IFN pathway in mediating the inflammatory response against MDR and virulent *A. baumannii* strains in an acute and severe sepsis model.

## Results

### The multidrug-resistant strain *A. baumannii* ATCC BAA-1605 activates caspase-1 and caspase-11 inflammasomes and induces programmed cell death

Previous work has reported that *A. baumannii* membrane proteins elicit an inflammatory response by inducing the expression of pro-inflammatory cytokines IL-1β and IL-18[25]. Indeed, we noted elevated IL-1β and IL-18 cytokine secretion levels in primary wild-type (WT) bone marrow-derived macrophages (BMDMs) infected with the multidrug-resistant (MDR) virulent *A. baumannii* ATCC BAA-1605 strain (hereafter named *A. baumannii* 1605) (Fig. 1a). To identify the inflammasome sensors responsible for the recognition of *A. baumannii*, we inoculated WT BMDMs with *A. baumannii* 1605 and measured *Nlrc4, Aim2, Nlrp3* and *Caspase-11* transcript levels in cell lysates at 6- and 12-hours post infection. We found a sustained high expression in *Nlrp3* and *Casp11* over time. These data suggest that *Nlrp3* and *Casp11* are potential sensors for *A. baumannii* 1605 (Fig. 1b). To further ascertain this finding, we inoculated *Nlrp3⁻/⁻*, *Asc⁻/⁻*, *Casp11⁻/⁻* and *Casp1/11⁻/⁻* BMDMs with *A. baumannii* 1605. We noted a strong reduction to an abolition of IL-1β secretion in all knockouts, and a significant decrease in IL-18 in *Casp11⁻/⁻* and *Casp1/11⁻/⁻* BMDMs. In contrast, the release of TNFα for BMDMs from all WT and knockout BMDMs

was maintained (Fig. 1a). These data suggest an activation of NLRP3 inflammasome via caspase-1 and/or caspase-11 pathways. Indeed, we found an activation of both caspase-1 and caspase-11 in WT and knockout BMDMs via immunoblotting (Fig. 1c). Activation of caspase-1 and caspase-11 cleaves the N-terminal end of Gasdermin D (GSDMD), resulting in a mature GSDMD, which forms membrane pores leading to pyroptosis[10]. We observed GSDMD cleavage in WT BMDMs (Fig. 1c), increased cell death (Fig. 1d). GSDMD cleavage was almost abolished in *Casp11⁻/⁻* and *Casp1/11⁻/⁻* BMDMs (Fig. 1c). Programmed cell death, was reduced in *Gsdmd⁻/⁻* BMDM compared to WT BMDM (Fig. 1d). Collectively these data indicate that *A. baumannii* 1605 strain induces both activation of caspase-1 and caspase-11 leading to pro-inflammatory cytokine secretion, Gasdermin D proteolytic cleavage, programmed cell death and activation of the NLRP3-Caspase-1 inflammasome.

### Virulent MDR *A. baumannii* 1605 infection predominantly activates caspase-11-NLRP3 inflammasome

To confirm whether *A. baumannii* 1605 activate both caspase-1 and caspase-11, we inoculated *Casp1⁻/⁻*, *Casp11⁻/⁻*, *Casp1/11⁻/⁻*, *Asc⁻/⁻* and *Nlrp3⁻/⁻* mice with *A. baumannii* 1605 intra-peritoneally at $2 \times 10^7$ CFU/mouse and measured survival, bacterial burden, and plasma IL-1β or IL-18 levels. Both *Nlrp3⁻/⁻* and *Asc⁻/⁻* mice were protected against *A. baumannii* 1605 (~60% survival rate) (Fig. 2a) due partly to a reduction of the bacterial burden and reduction of circulating pro-inflammatory cytokines (Fig. 2b–d). Remarkably, we found 80% survival of the infection (Fig. 2a), a significant reduction of the bacteria burden (Fig. 2b, c) and reduction of the plasma pro-inflammatory cytokine secretion (Fig. 2d) in *Casp11⁻/⁻* mice. Although *Casp1⁻/⁻* and *Casp11⁻/⁻* (Fig. 2a) exhibited a similar survival rate, we found that *Casp1⁻/⁻* displayed a similar bacterial burden but lower circulating pro-inflammatory cytokine levels compared to WT mice (Fig. 2b-d). Interestingly, *Casp1/11⁻/⁻* mice exhibited a similar survival rate to WT mice (~20% survival; Supplementary Fig 2a) with a similar bacterial burden to WT (Supplementary Fig. 2b, c) and significantly lower levels of circulating pro-inflammatory cytokines (Supplementary Fig. 2d), suggesting a lack of protective cytokine production in response to bacteriemia. We next examined the role of GSDMD-mediated cell death by infecting *Gsdmd⁻/⁻* mice. *A. baumannii* 1605 bacteria inoculation in *Gsdmd⁻/⁻* mice resulted in a slight increase in survival (60% survival rate) when compared to WT mice (Supplementary Fig. 3c, d). While the bacterial burden in the *Gsdmd⁻/⁻* mice did not differ from WT mice, we observed a 10-fold reduction in plasmatic IL-18 in *Gsdmd⁻/⁻* mice (Supplementary Fig. 3b) in agreement with a recent report[26]. We similarly found a 10-fold reduction in IL-18 level in the lysates of infected *Gsdmd⁻/⁻* BMDM, and a marginal decrease in IL-1β and TNFα secretion level (Supplementary Fig. 3a). Taken together it suggests *A. baumannii* 1605 activates both caspase-1 and caspase-11 resulting in the activation of the NLRP3/ASC inflammasome. Additionally, these data confirmed that while NLRP3/ASC, caspase-1 or caspase-11 deficiency alone conferred mouse survival whereas deficiency in both caspase-1 and caspase-11 did not, possibly due to the overall reduction of IL-1β or IL-18 and increase in TNFα secretion in caspase 1/11 deficiency.

### Reduced programmed cell death in the lung partly underlies host resistance to *A. baumannii* 1605 infection

We next sought to determine how *Casp1⁻/⁻*, *Casp11⁻/⁻*, *Nlrp3⁻/⁻* and *Asc⁻/⁻* mice were protected against the infection, whereas *Casp1/11⁻/⁻* was not. We hypothesised that protection against the infection requires recruitment of neutrophils and inflammatory monocytes to clear the bacteria from the infected tissues. Previous works have reported that during lung infection, depletion of neutrophils resulted in an acute lethal infection[27], and macrophage depletion lead to an increased tissue bacterial burden[28]. We postulated that the first line of cellular effectors against acute bacterial infection and sepsis to produce type I interferon cytokines and exert their phagocytic functions are neutrophils (CD11b⁺Ly6g⁺) and inflammatory monocytes (CD11b⁺Ly6c⁺)[29]. We hypothesise these immune cell effectors are required

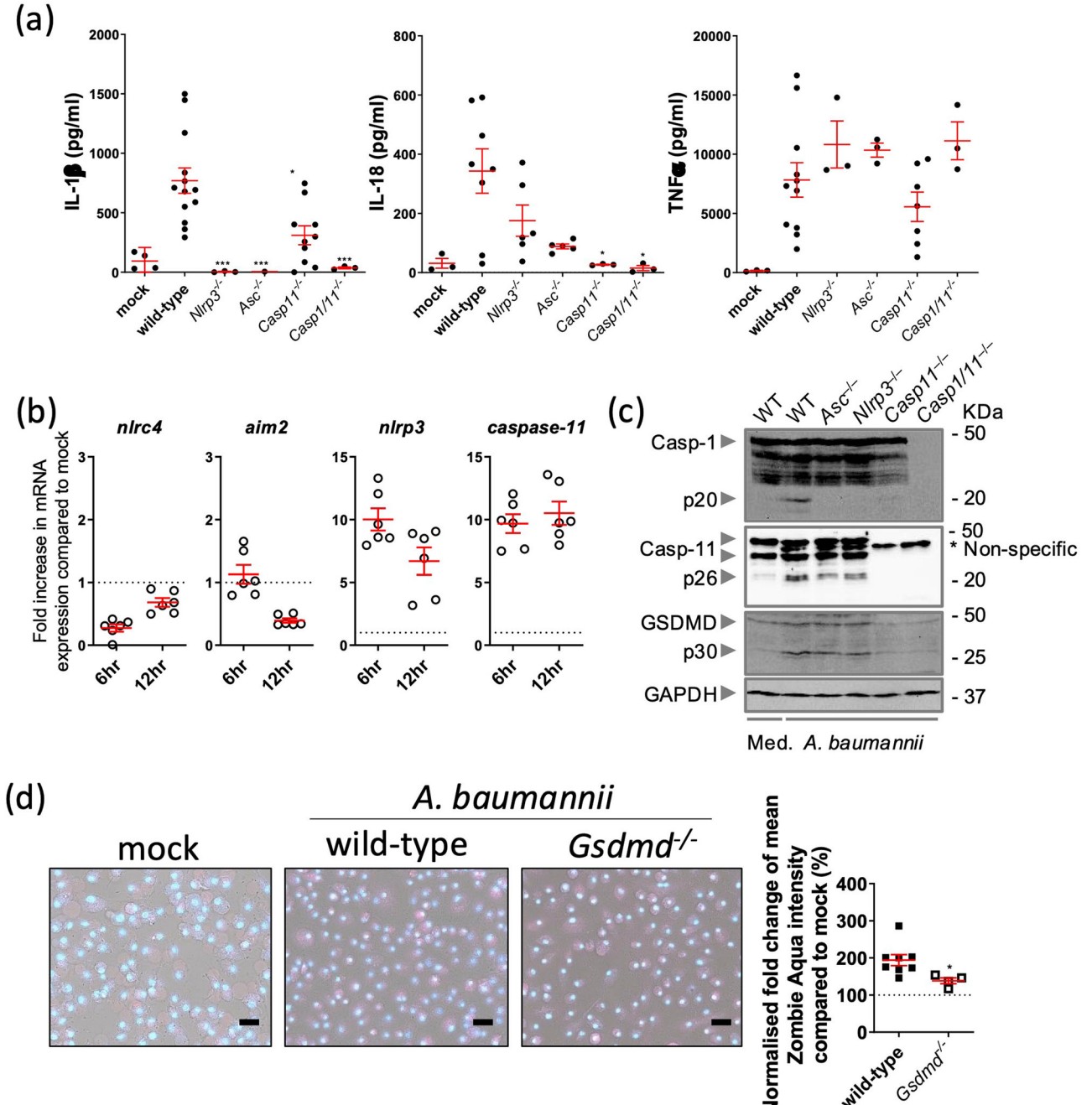

**Fig. 1 | *A. baumannii 1605* bacteria induce inflammasome activation. a** Cytokine levels IL-1β, IL-18 and TNFα in supernatants from Mock, WT and mutant mouse BMDMs 12 h post infection (MOI = 10), *n* = 3–13 biological replicates, each dot represents one replicate. **b** Transcript levels of inflammasome sensor genes *Nlrc4, Aim2, Nlrp3* and *Caspase-11* produced in wild-type mouse BMDMs 6 h post infection (MOI = 10) and measured by quantitative PCR, *n* = 6, mean ± SEM. **c** Western blots on activated caspases and GSDMD 16 h post infection (MOI = 10). **d** Immunofluorescence imaging of BMDMs cell death post 24 h of infection (MOI = 10). Red: zombie aqua, blue: Hoechst. Scale bar 30 μm. *$P < 0.05$, **$P < 0.01$, ***$P < 0.001$ compared to wild-type. mean ± SEM. Non-parametric *t*-test was used to compare differences between groups.

for bacterial clearance by increase recruitment and/or increased clearance of infected cells in the target tissues, such as the lung, were responsible for enhanced resistance to *A. baumannii* 1605. While we indeed confirmed - by flow cytometry - an increase of CD11b⁺Ly6g⁺ and CD11b⁺Ly6c⁺ populations in the lungs of infected WT mice up to 20-hour post inoculation (Supplementary Fig. 4a), we observed no significant difference in the percentage of CD11b⁺Ly6g⁺ and CD11b⁺Ly6c⁺ in the lungs between WT, *Asc*⁻/⁻, *Casp1/11*⁻/⁻ and *Casp11*⁻/⁻ mice but noticed a reduction of cell numbers in *Casp1/11*⁻/⁻ mice (Fig. 3a, b). Interestingly, we noted a reduction in chemokine *Cxcl1* expression level, markers of neutrophils recruitment in

*Nlrp3*⁻/⁻ and *Asc*⁻/⁻ mice (Fig. 3c) while the expression levels of the neutrophil chemokine receptors *Cxcr1* and *Cxcr2* remained similar in the lungs between the WT and the four knockout strains (Fig. 3c). There was a significant decrease in the expression of the inflammatory monocyte marker, *Ccr2*, in *Casp1/11*⁻/⁻ mice, consistent with the lack of inflammation (Supplementary Fig 2d). However, we observed no difference between the WT and knockout mice in the expression of two other markers of inflammatory monocyte recruitment *Ccl2* and *Ccr5* (Fig. 3d). We finally observed no difference in the pathology of the lungs (Supplementary Fig. 4b–d). Taken together, these findings suggest that increased neutrophils and

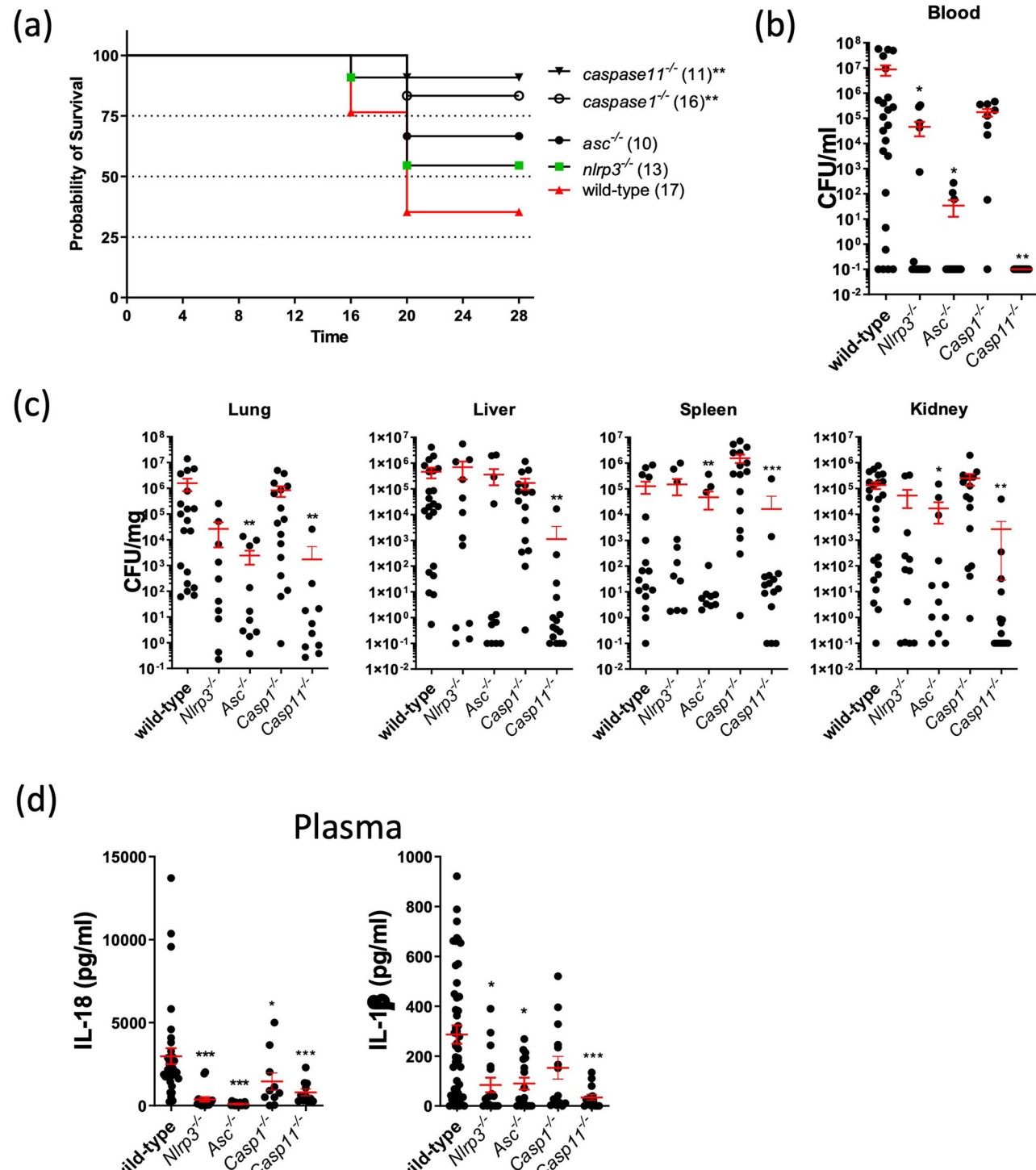

**Fig. 2 | Inflammasome deficient mice confer resistance to *A. baumannii* 1605 infection. a** Survival rate of WT, *Nlrp3*$^{-/-}$, *Asc*$^{-/-}$, *Caspase1*$^{-/-}$ and *Caspase11*$^{-/-}$ mice 28 h post infection (i.p. 2×10$^7$ CFU/mouse). **b** The bacteriemia and (**c**) bacteria dissemination to different organs (lung, liver, spleen or kidney) at 10–20 h post infection were quantified by serial dilution on trypticase soy broth and CFU counting. **d** Cytokine levels IL-1β and IL-18 in plasma 12 h post infection. Data were collected from at least three independent experiments, numbers of mice (*n*) are indicated in parentheses, *$P < 0.05$, **$P < 0.01$, ***$P < 0.001$ compared to wild-type. mean ± SEM. Kaplan–Meier estimate was used to compare mice survival rates. Non-parametric *t*-test was used to compare differences between groups.

inflammatory monocytes recruitment are unlikely to play a major role in the survival of *Asc*$^{-/-}$ and *Casp11*$^{-/-}$ mice.

We next reasoned that a reduction in neutrophil/monocyte cell death might instead have protected *Nlrp3*$^{-/-}$, *Asc*$^{-/-}$ and *Casp11*$^{-/-}$ mice against deleterious inflammation and led to a reduction in bacteria burden. We quantified the percentage of neutrophils undergoing programmed cell death in the lungs of these mice post-infection using Zombie aqua dye. All knock-out mouse strains showed a significant decrease in the number of dead or dying neutrophils including *Casp1/11*$^{-/-}$ mice (Fig. 4a). Remarkably, while we found no difference in Zombie aqua-positive inflammatory monocytes between WT and *Casp1/11*$^{-/-}$ mice, we observed significantly lower proportion of Zombie aqua-positive cells for *Asc*$^{-/-}$ and *Casp11*$^{-/-}$ mice

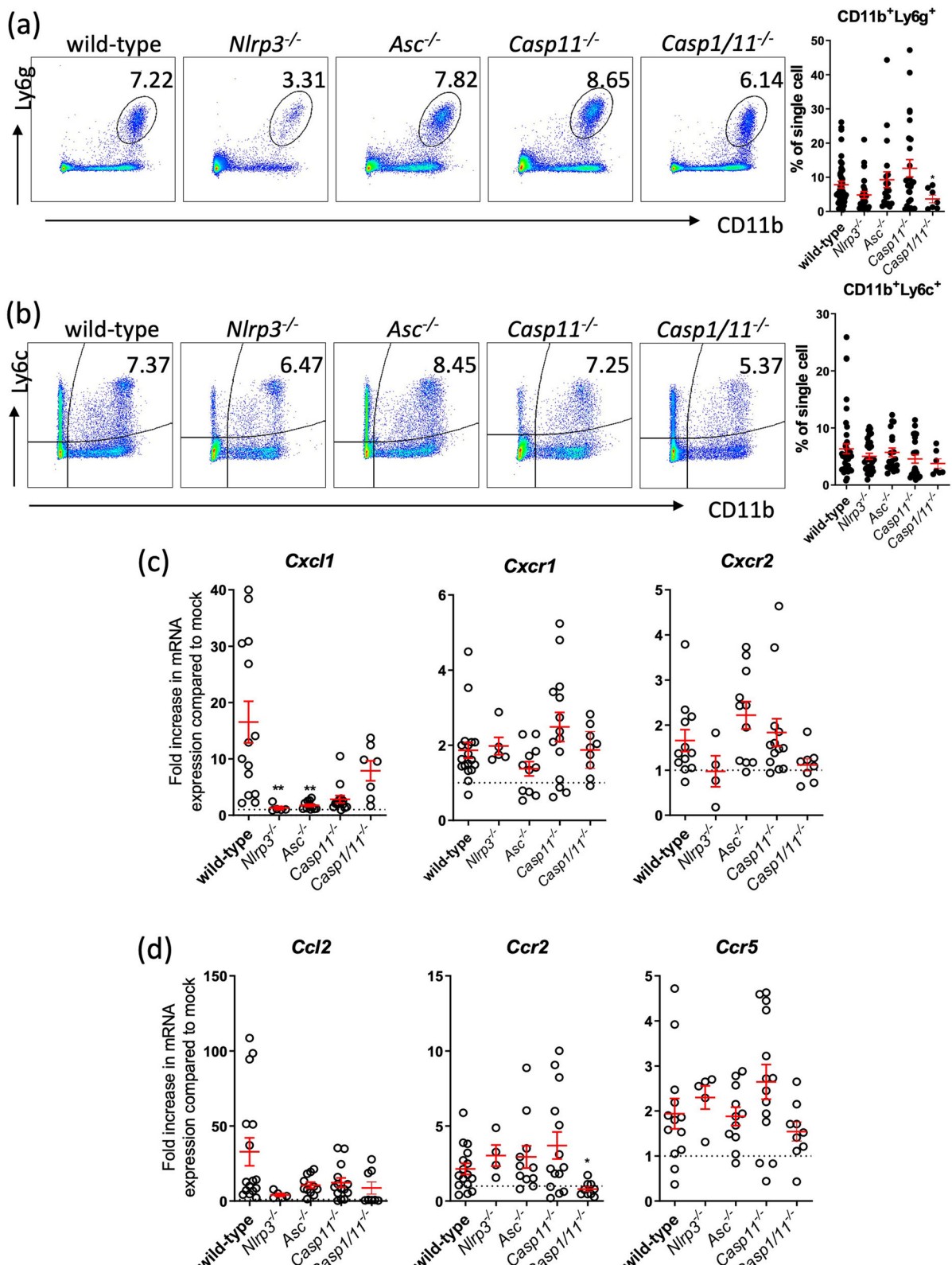

**Fig. 3 | Absence of NLRP3 signalling decreases neutrophil recruitment to the lungs.** Flow cytometry quantification of (**a**) percentage of neutrophils (CD11b⁺Ly6g⁺) and (**b**) inflammatory monocytes (CD11b⁺Ly6c⁺Ly6g⁻), in the mice lung 14–20 h infection (i.p. $2 \times 10^7$ CFU/mouse), $n = 20$. qPCR quantification of induction of (**c**) neutrophil chemokine and chemokine receptors or (**d**) inflammatory monocyte chemokine and receptors. Data were collected from at least three independent experiments, $n = 4–14$ for each group, each data point represents a replicate. *$P < 0.05$, mean ± SEM. Non-parametric $t$-test was used to compare differences between groups.

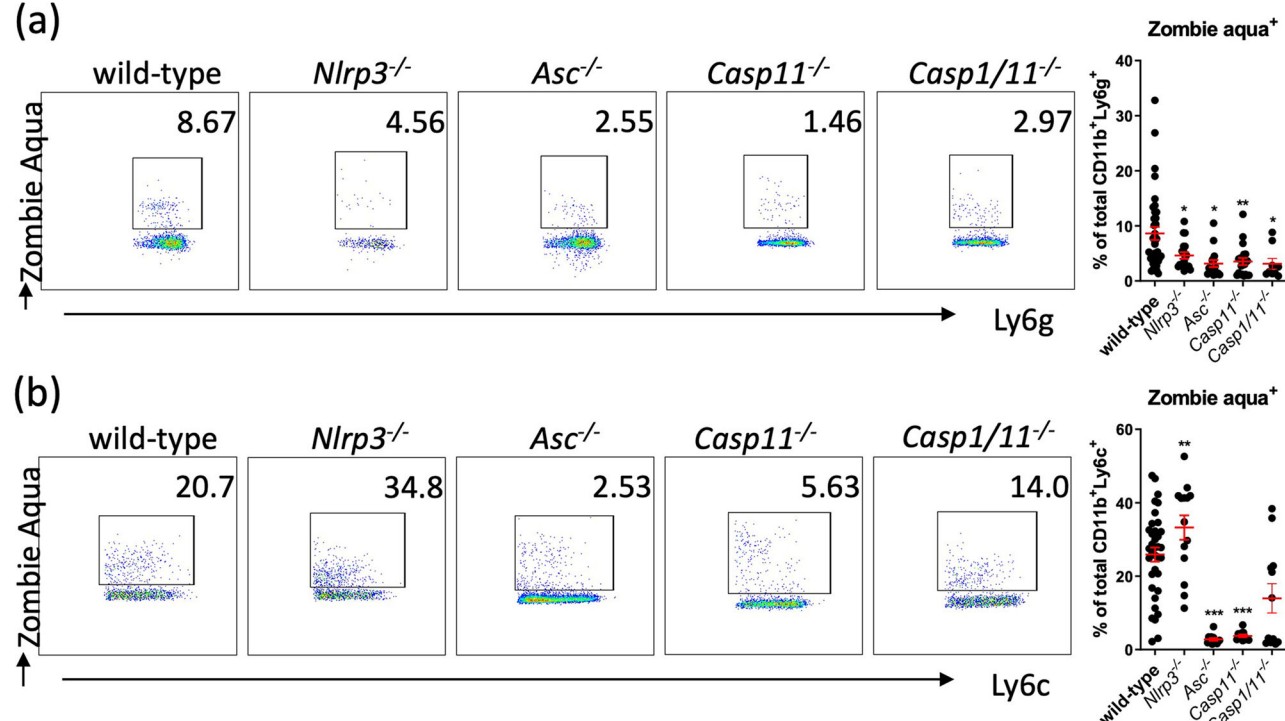

**Fig. 4 | Absence of inflammasome signalling decreases effector cell death.** Flow cytometry quantification of (**a**) neutrophils (CD11b+Ly6g+) cell death (Zombie Aqua+) and (**b**) inflammatory monocytes (CD11b+Ly6c+Ly6g-) cell death, in mice lung 14–20 h infection (i.p. $2 \times 10^7$ CFU/mouse). *$P < 0.05$, **$P < 0.01$, ***$P < 0.001$, mean ± SEM, $n = 20$. Non-parametric $t$-test was used to compare differences between groups.

suggesting a reduced inflammatory monocyte cell death in these strains from WT mice (Fig. 4b).

These data collectively suggest that in $Nlrp3^{-/-}$, $Asc^{-/-}$, $Casp11^{-/-}$ and $Casp1/11^{-/-}$ mice, the balance between effector recruitment (increase cell recruitment) and programmed cell death (lower cell death) might have contributed to lower bacterial burden and decreased host mortality.

### Type I IFN is required for host protection and bactericidal activity against MDR *A. baumannii* 1605

A previous study and our group have reported that activation of caspase-11 in *A. baumannii* infection is dependent on type I IFN signalling[30,31]. We reasoned that type I IFN priming is potentially required in *A. baumannii* 1605-induced caspase-11 non-canonical pathway activation resulting in detrimental inflammation. To investigate the contribution of type I IFN signalling to inflammasome activation by *A. baumannii*, we inoculated WT and $Ifnar^{-/-}$ BMDMs with *A. baumannii* 1605 and measured $Nlrp3$ and $Caspase-11$ transcript levels and pro-inflammatory cytokine levels up to 12 h post infection. We observed a reduction in $Caspase-11$ transcript levels while $Nlrp3$ transcript levels were unchanged in $Ifnar^{-/-}$ BMDMs (compared to WT) (Fig. 5b). Additionally, we noted a reduction in pro-inflammatory cytokine expression and secretion (Fig. 5a and Supplementary Fig. 5) in $Ifnar^{-/-}$ BMDMs suggesting that IFN primes caspase-11 activation, 12 h post *A. baumannii* 1605 infection. Next, we inoculated WT and $Ifnar^{-/-}$ mice intraperitoneally with *A. baumannii* 1605 bacteria at $2\times10^7$ CFU/mouse. We assessed survival, bacterial load and plasmatic pro-inflammatory cytokine secretion. We found a protection of the $Ifnar^{-/-}$ mice with > 50% survival rate 28 h post inoculation (Fig. 5c). Interestingly, we observed a significant reduction in bacterial load in the lungs but not in the other organs (Fig. 5d, e) and quasi abolition of the pro-inflammatory plasma cytokine levels in $Ifnar^{-/-}$ mice (Fig. 5f). These results suggest a protective role when type I IFN signalling is abolished during infection with MDR *A. baumannii* 1605. We next sought to determine the mechanism/s of resistance of the $Ifnar^{-/-}$ mice. We analysed CD11b+Ly6g+ and

CD11b+Ly6c+ populations, chemokine receptor levels and Zombie aqua-positive cells in the lungs of $Ifnar^{-/-}$ and WT mice, 16-20 h post infection. We found a significant reduction in CD11b+Ly6g+ population (Supplementary Fig. 6a) but not CD11b+Ly6c+ (Supplementary Fig. 7a) and a substantial reduction in Zombie aqua-positive cells in CD11b+Ly6g+ and CD11b+Ly6c+ cell types in $Ifnar^{-/-}$ compared to WT mice (Supplementary Figs. 6c and 7c). Interestingly, we found no difference in chemokine levels (Supplementary Figs. 6b and 7b). Together these findings confirm that an IFN-dependent caspase-11 response is required following MDR *A. baumannii* 1605 infection to mediate the release of pro-inflammatory cytokines and encourage the persistence of effector cells in target organs.

### Deficiency in IFN-inducible guanylate binding protein 1 protects the host against *A. baumannii* via caspase-11 rather than direct killing

The IFN-inducible guanylate-binding protein – human GBP1 - is a cytosolic receptor for LPS and triggers pyroptosis during infection with certain Gram-negative bacteria, such as *Salmonella Typhimurium*[20,22] and *Legionella pneumophila*[32]. Human GBP1 binds directly to cytoplasmic LPS, recruits other GBPs and promotes an oligomerization state of caspase-4[33]. Therefore, human GBP1 orchestrates the recruitment of other GBPs and caspase activation underscoring its central role in cytosolic host protection against pathogens. Additionally, phosphate groups on the lipid A of LPS play an essential role in promoting GBP1-LPS interaction and activation of the non-canonical inflammasome pathway[20]. In *A. baumannii* infection, Colistin resistance to antibiotic therapy is mediated by LOS via direct binding to lipid A (LpxA)[24]. We speculated in the context of *A. baumannii* infection that mouse GBP1 might induce caspase-11-dependent pyroptosis via LOS-dependent killing of *A. baumannii*. We generated the mouse $Gbp2b^{-/-}$ (thereafter named its synonym $Gbp1^{-/-}$) knockout strain in mice using CRISPR-Cas9 gene editing technology[21]. We then assessed survival, bacterial burden, inflammasome activation and pyroptosis. We observed that $Gbp1^{-/-}$ mice were highly protected against *A. baumannii* 1605

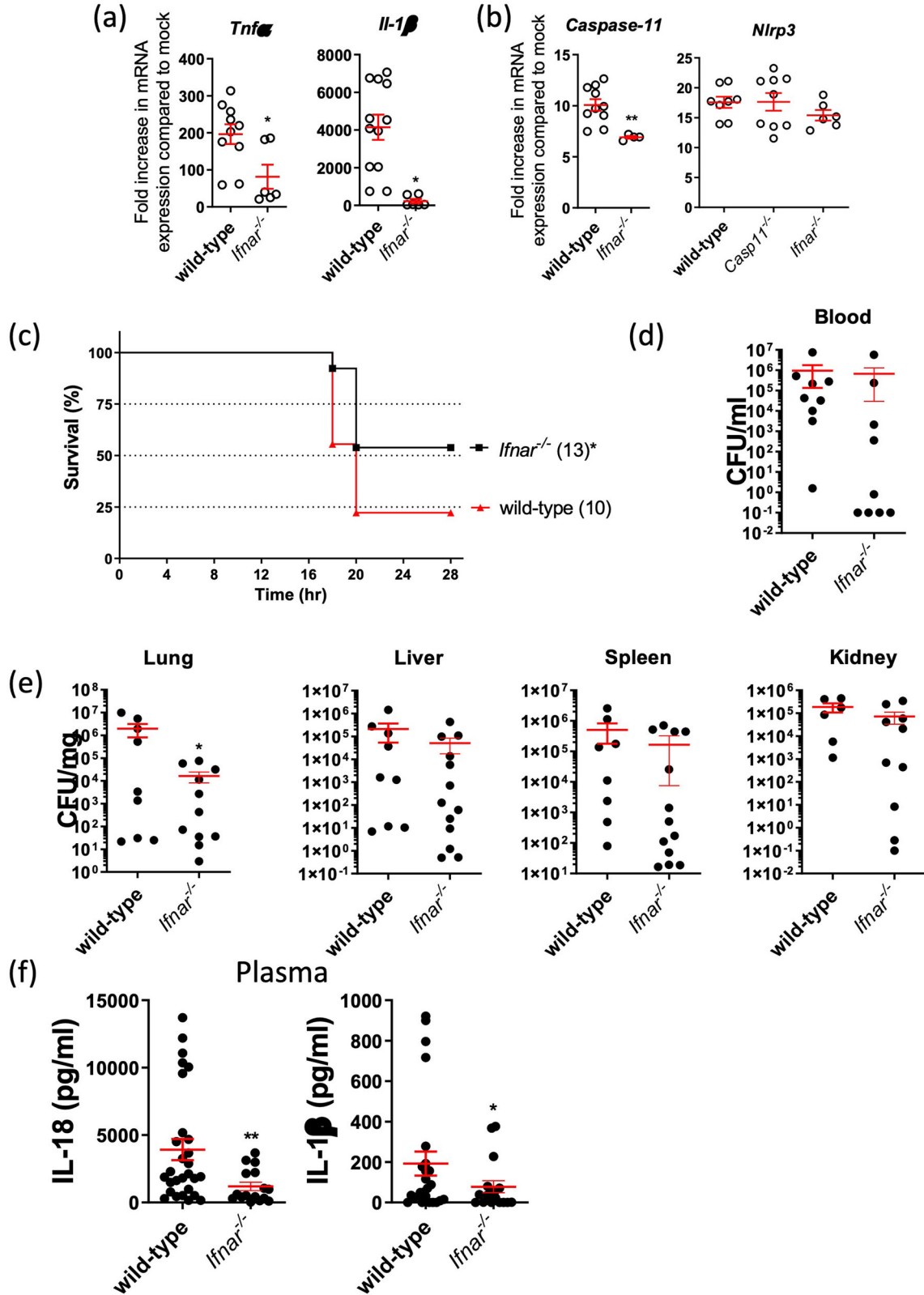

**Fig. 5 | *A. baumannii* induces type I IFN-dependent inflammasome activation.** Transcript levels measured by quantitative PCR of (**a**) inflammatory cytokines and (**b**) inflammasome sensors in mouse BMDMs 6 h after infection (MOI = 10), $n = 6$–12, each data point represents a replicate. **c** Mice survival rate, (**d**) the level of bacteraemia, (**e**) bacteria dissemination to different organs and (**f**) plasma cytokine levels 16–20 h post *A. baumannii* 1605 infection (i.p. $2 \times 10^7$ CFU/mouse), numbers of mice (*n*) are indicated in parentheses. Data were collected from at least three independent experiments, number of biological samples (*n*) as indicated in parentheses, *$P < 0.05$, **$P < 0.01$ compared to wild-type. mean ± SEM. Kaplan–Meier estimate was used to compare mice survival rates. Non-parametric *t*-test was used to compare differences between groups.

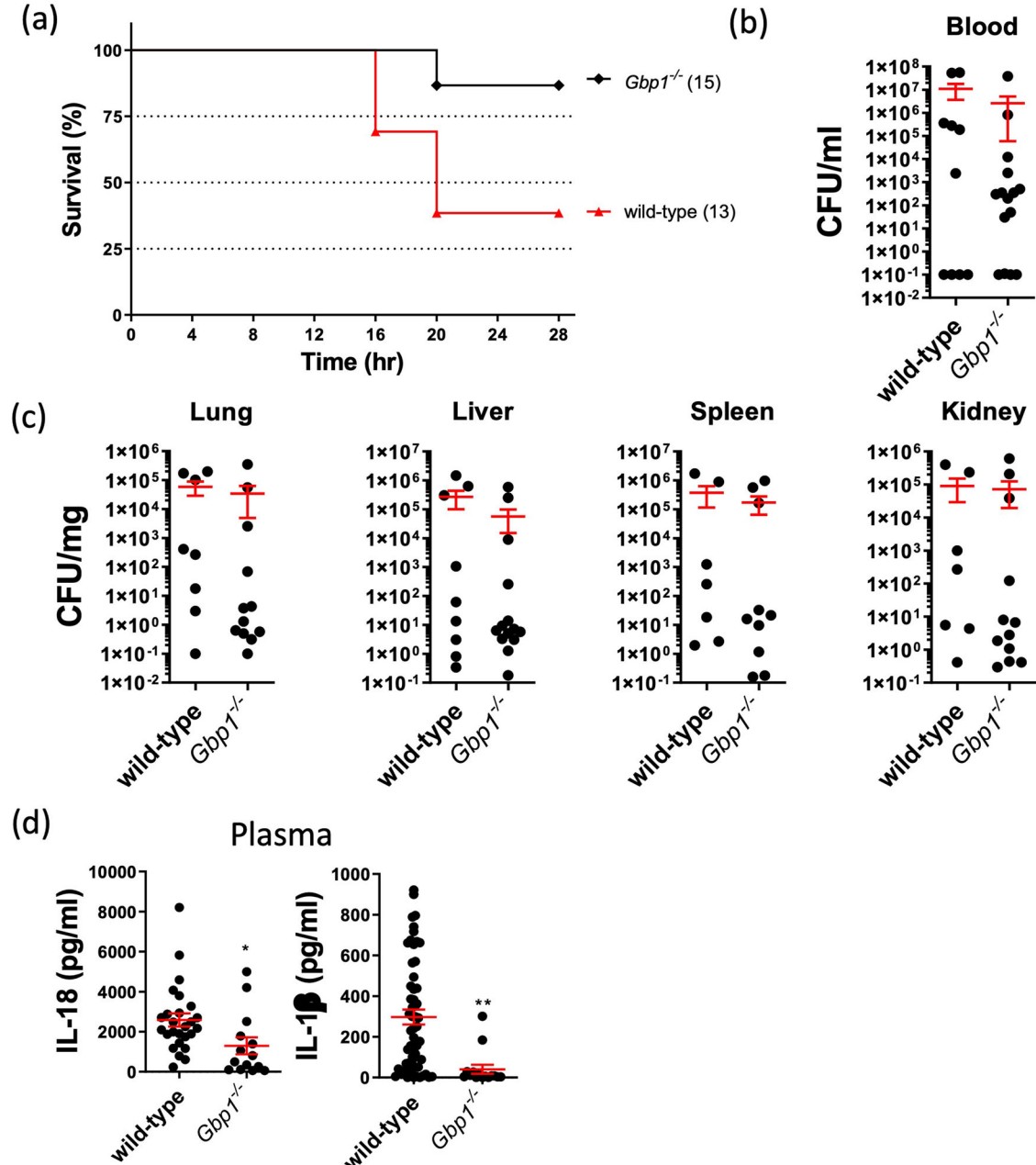

**Fig. 6 | GBP1 drives acute lethality in *A. baumannii*-infected mice. a** Mice survival rate, (**b**) the level of bacteriemia, (**c**) bacteria dissemination to different organs and (**d**) plasma cytokine levels 16–20 h post *A. baumannii* 1605 infection (i.p. $2 \times 10^7$ CFU/mouse). Data were collected from at least three independent experiments, numbers of mice (*n*) are indicated in parentheses, \**P* < 0.05, \*\**P* < 0.01 compared to wild-type. mean ± SEM. Kaplan–Meier estimate was used to compare mice survival rates. Non-parametric t-test was used to compare differences between groups.

infection with > 70% survival rate, although with no reduction in bacterial burden (Fig. 6a–c) but a strong reduction in plasma pro-inflammatory cytokine levels (Fig. 6d). Immunoblotting confirmed the activation of caspase-11 was impaired and GSDMD proteolytic cleavage reduced in *Gbp1*⁻/⁻ BMDMs compared to WT BMDMs (Fig. 7a). Additionally, we found a strong reduction in pro-inflammatory cytokine levels at 12 h post-inoculation (Fig. 7d). These data suggest that mouse GBP1 mediates caspase-11 inflammasome activation in response to *A. baumannii* 1605.

Next, to assess the role of LpxA in GBP1-mediated killing, we infected WT and *Gbp1*⁻/⁻ BMDMs with an *A. baumannii* LOS-deficient strain carrying a nonsense mutation in *LpxA* (19606 R) and its complement strains (19606R + LpxA or AL 1847, 19606 R + V) and *A. baumannii* ATCC 19606 (WT 19606) and 1605 strain as controls[24]. We found in BMDMs,

either 19606 R or their complement strains did not alter significantly the inflammatory response from WT 19606 or *A. baumannii* 1605 strains in WT and *Gbp1*⁻/⁻ BMDMs (Fig. 7a, b, d–f). We next assessed direct killing from mouse GBP1 by co-incubating the full-length purified mouse GBP1 protein with *A. baumannii* strains. Interestingly, we found the full-length purified mouse GBP1 protein exerted a bactericidal activity against *A. baumannii* strains in a dose-dependent manner with an IC50 between 40 to 80 µg/ml, which corresponds to non-physiological concentrations (Supplementary Fig 8). Again, no significant change was observed between 1605, 19606 WT, 19606R and 19606R+LpxA (Fig. 7c). Together, these results suggest that *Gbp1*⁻/⁻ protects the host against *A. baumannii* via activation of the caspase-11-NLRP3 pathway and GSDMD proteolytic cleavage rather than direct killing *A. baumannii* bacteria.

**Fig. 7 | GBP1 drives LPS-independent *A. bau-mannii* responses.** Representative western blots on activated caspases and GSDMD of *A. baumannii* (**a**) *A. baumannii* 1605 and the *LpxA* deficient A1847 strains at 16 h post-infection (MOI = 10).
**b** Cytokine IL-1β, IL-18, IL-6 and CXCL1 levels in supernatants 12 h post infection (MOI = 10), *n* = 3–7, each data point represents a replicate.
**c** TNFα levels post different *A. baumannii* strains infection, *n* = 4. \*P < 0.05, \*\*P < 0.01, \*\*\*P < 0.001 compared to the respective wild-type. mean ± SEM. Non-parametric *t*-test was used to compare differences between groups.

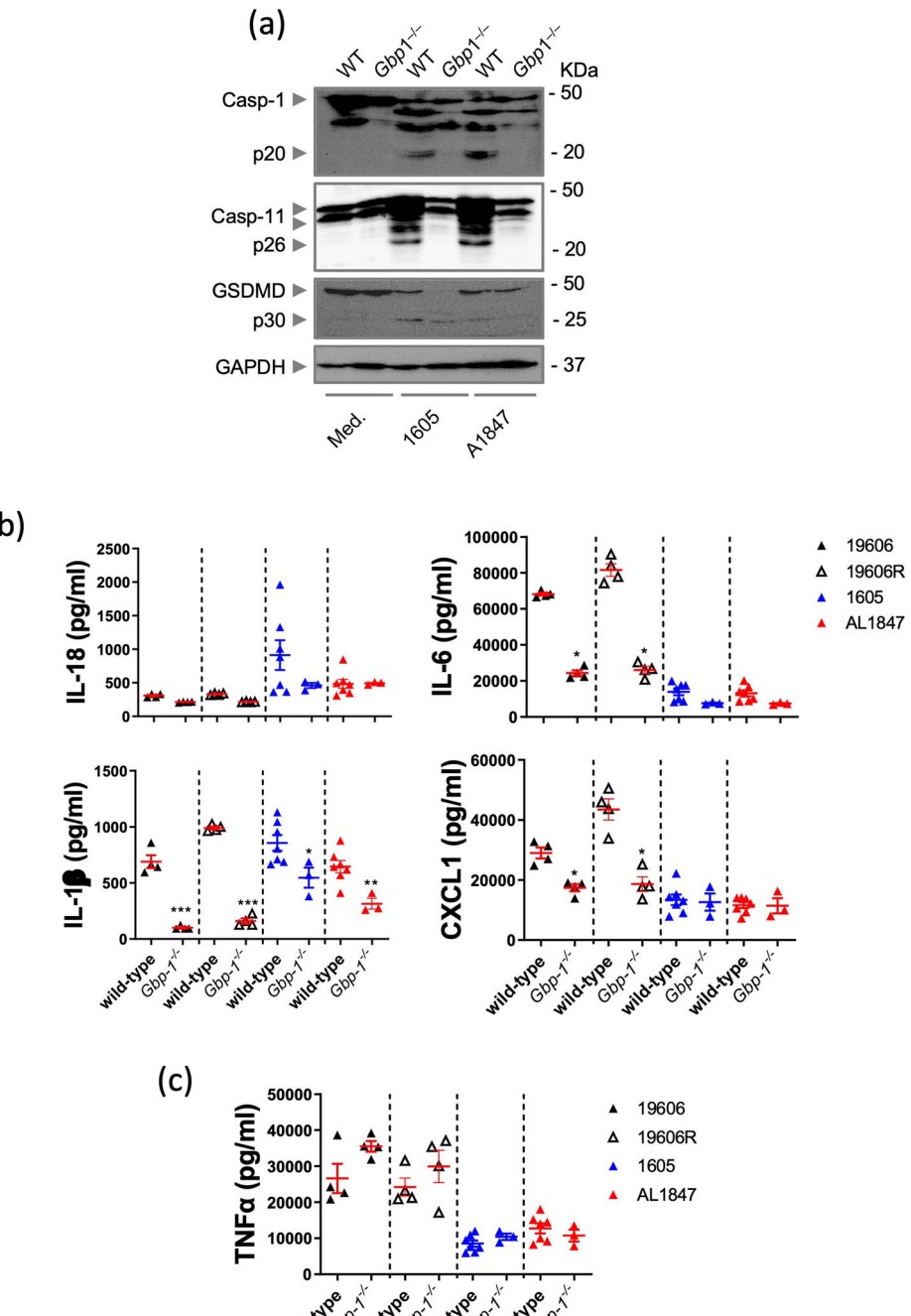

## Discussion

*A. baumannii* is a Gram-negative bacterium that causes opportunistic pulmonary and systemic pathologies in humans and is of importance for its resistance to last-resort antibiotics[3]. Despite the clinical significance of *A. baumannii* in humans, little is known about the role of the innate immune system in host defence against the pathogen. Inflammasome activation is key for innate immune recognition of pathogens and for innate host defences[34]. While NLRP3 activation has been reported as critical for host immunity in response to *A. baumannii*[15,35,36], the role of non-canonical inflammasome activation and cytosolic immunity in sensing *A. baumannii* bacteria remained elusive and poorly understood. A deeper understanding of the host immune defence mechanisms is required to devise potential novel therapies against MDR/XDR bacteria.

Our findings demonstrated that recognition of MDR *A. baumannii* 1605 infection by the host activates the caspase-1 and caspase-11, resulting in NLRP3/ASC inflammasome activation and the formation of GSDMD pores and induction of programmed cell death. Importantly, we found that caspase-11, via type I IFN, regulates the tight balance between protective and deleterious inflammation. Finally, we discovered that upon bacterial recognition, the cytosolic molecule GBP1 exerts a protective effect by likely activating caspase-11 inflammasome rather than by direct killing. Together our findings have demonstrated that innate immune defences mediated by caspase-11, IFN and GBPs are required to facilitate protective inflammation against MDR *A. baumannii* bacteria (Fig. 8).

Previous reports have identified the requirement of NLRP3/ASC and the caspase-1 canonical pathway[15,35] as well as a protective role of caspase-11

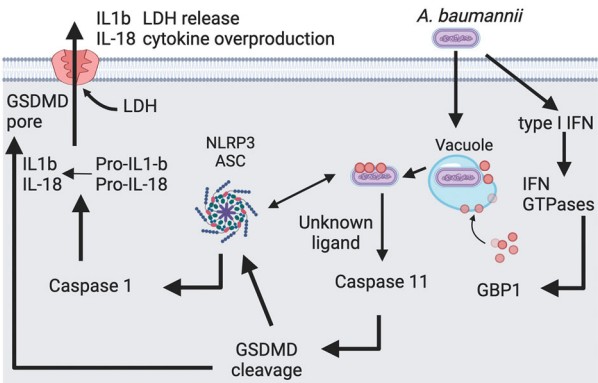

**Fig. 8 | Proposed model of the non-canonical inflammasome response to *A. baumannii* 1605.** *A baumannii* infects a mammalian cell and triggers type I IFNa response, which leads to GBP1 production and binding to the bacteria in the vacuole. It triggers a Caspase 11 activation and GSDMD-mediated pyroptosis. In parallel the canonical activation via Caspase 1 leads to LDH release and cytokine over-production. (drawn with BioRender).

in the host inflammatory response to *A. baumannii*[37]. Our data confirm these findings and establish that NLRP3 inflammasome is activated by MDR *A. baumannii* 1605. Our findings however differ from Wang and colleagues[37] likely due to the mode of administration and the bacteria concentration used in our acute and severe infection model. We further investigated the mechanisms of host protection and the role of caspase-11 and GSDMD dependency during MDR *A. baumannii* 1605 infection. In agreement with a previous report[30], we found MDR *A. baumannii* 1605 triggers caspase-11 and GSDMD-induced cell death. The virulence of the bacterial strain outer membrane, vesicle/capsule composition, the mode of infection and the bacterial concentration differ from this previous report, which has resulted in a differently observed outcome. While IL-1β and IL-18 pro-inflammatory cytokines secretion were reduced or abolished in caspase knockout mice, other pro-inflammatory inflammasome independent cytokines secretion (TNFα and IL-6) were maintained or elevated. It suggests that a balance between a lower to abolished inflammasome-dependent pro-inflammatory cytokine secretion and elevated TNFα and IL-6 secretion is likely causative for caspase 1/11 double knockout death. However, it is possible that an additional distinct pathway might explain the lack of protective effect for caspase 1/11. Our findings and our previous assessment on multiple strains with different growth rates and inflammatory responses[31], however, establish that the non-canonical inflammasome activation via caspase-11-mediated cell death is major host defence mechanisms against MDR *A. baumannii*.

Neutrophils and inflammatory monocytes are abundant resident populations and are rapidly recruited to the infection site to kill micro-organisms[38]. Previous studies have reported that early recruitment of neutrophils to the lungs was mediated by NLRP3 and was dispensable for the host defence[27,39–41]. Our studies revealed that in addition to early recruitment to the target tissues, a balance between effector cell recruitment and their persistence is important to maintain the neutrophils/monocytes effector cells in the tissues. This balance might be partly responsible for modulating the host resistance against MDR *A. baumannii*. It therefore suggests the innate immune response against *A. baumannii* infection might be driven by a tight regulation between effector cell recruitment and programmed cell death mediated by inflammasome activation, controlled by NLRP3/ASC and caspase-11.

Previous work identified IFN is required for host resistance to virulent *A. baumannii* infection by mediating multiple cell death pathways via caspase-11[30]. However, the role of cytosolic immunity mediated by IFN molecules for *A. baumannii*-mediated infection is unknown. We did, however, demonstrate the requirement of IFN in mediating the immune response via caspase-11 activation, cell death and the activation of IFN induced bactericidal proteins in agreement with a previous report[30].

Importantly, our findings highlight the requirement of GBP1 in the response to MDR *A. baumannii* 1605. Previous work identified human GBP1 as cytosolic receptors for LPS in *S. Typhimurium* and *S. flexeneri*[20,42,43] while it was not demonstrated for the mouse GBP1[21]. Here, we first demonstrated that full-length purified GBP1 exerts a non-physiological bactericidal activity against LOS-deficient strains resistant to polymyxin, as well as a MDR *A. baumannii* 1605 strain resistant to carbapenems. Unlike human GBP1, our data, therefore, suggest that the mouse GBP1 mechanism of resistance does not result in a direct binding of human GBP1 to LPS[20,33]. Our finding is in line with previous studies on *Neisseria meningitidis*[21] and *Moraxella catarrhalis* LOS[44] and *E. coli* LPS[45], demonstrating no direct interaction between mouse GBP1 and the LOS. Importantly our results rather demonstrate the role of GBP1 against *A. baumanii* in mice by activating the non-canonical inflammasome pathway.

In conclusion, we demonstrated a critical role of the caspase-1 and caspase-11 non-canonical pathways and type I IFN pathways in mediating bacterial killing, activating the inflammasome and protective pro-inflammatory cytokine production against *A. baumannii*.

## Method
### Bacteria strains
*Acinetobacter baumannii* BAA-1605 strain (*A. baumannii* 1605) was obtained from the American Type Culture Collection (ATCC). This strain is a multi-drug resistant to many antibiotics including carbapenems (resistant to ceftazidime, gentamicin, ticarcillin, piperacillin, aztreonam, cefepime, ciprofloxacin, imipenem and meropemem). This stain was an isolate from sputum of military personnel returning from Afghanistan. *Acinetobacter baumannii* AL 1847, a clinical isolate harbouring a 30 bp mutation in *LpxA* resulting in a frameshift mutation, *Acinetobacter baumannii* 19606, AL 1847, an ATCC 19606 derivative harbouring a 30 bp mutation in *LxpA* resulting in a frameshift mutation, 19606R (resistant to polymyxin B, carrying a nonsense mutation in the *LpxA* gene) and their complemented strains 19606R-LpxA, 19606R-V (transformed with the empty shuttle vector pWH1266) were obtained from Monash University in Australia (Prof John Boyce) and were previously described[24]. Frozen stocks of bacteria were streaked onto trypticase soy agar plates (Cat. No.: 211768, BD, with 1.5% agar (BD Cat. No.: 281230)) and incubated overnight under aerobic conditions at 37 °C. Single colonies were picked and inoculated into 5 ml of trypticase soy broth and were incubated under aerobic conditions at 37 °C on an orbital shaker at 250 rpm for 16-18 h until cloudy. The bacteria were then sub-cultured in 30 ml of trypticase soy broth for further propagation for 4 h at 37 °C, on an orbital shaker at 250 rpm. The bacteria 19606 R and 19606 R+LpxA were maintained under colistin selection pressure[24]. Bacterial stocks were prepared by adding 30% of sterile glycerol (Cat. No.: G2025, Sigma-Aldrich), aliquoted into 2 ml vials and frozen at -80 °C before use.

### Primary bone marrow-derived macrophages
Mouse bone marrow derived macrophages were used as a mouse macrophage infection model. Total bone marrow was extracted from both mouse femurs, passed through a 70 µm cell strainer (Cat No.:352350, BD Falcon) and centrifuged at 430 relative centrifugal force (rcf) for 5 min. The supernatant was discarded. Red blood cells were lysed using 10 ml of 1x red blood cell (RBC) lysis buffer per mouse for 5 min during centrifugation.

To promote differentiation into macrophages, bone marrow cells were washed and seeded at $5\times10^6$/dish in 10 cm sterile dish in 10 ml of RPMI/10% Foetal Calf Serum (FCS) including 10 ng/ml of mGM-CSF (Cat. No.: 130-095-739, Miltenyi Biotech). Cells were incubated at 37 °C 5% $CO_2$, 95% relative humidity. The day of isolation is considered as day 0. On day 3, an extra 5 ml of fresh RPMI/10% FCS including 10 ng/ml of mouse GM-CSF was added to replenish the cytokines. On day 6 or 7, adherent cells were collected using RPMI/5 µM EDTA to detach cells from the plates. Cells were seeded in 96-well plates at $1\times10^5$/well in RPMI/10% FCS and incubated overnight at 37 °C 5% $CO_2$, 95% relative humidity, prior to infection.

## Immunofluorescence

BMDMs were seeded at $4 \times 10^5$/well in sterile 24-well glass bottom plate (Cat. No.: P24-1.5HN, Cellvis) prior to *A. baumannii* inoculation. *A. baumannii* strain 1605 was prepared at multiplicity of infection (m.o.i.) 10 and cells were infected for 24 h (final volume 1 ml/well) at 37 °C 5% $CO_2$ in air. Post infection, cells were washed twice with sterile 1xPBS before staining with Zombie Aqua (1:100, 100 µl/sample, Cat. No.: 423101, BioLegend) and Hoechst 33342 (80 micromole (µM)/100 µl/sample, Cat No.: H1399, Invitrogen, Carlsbad, CA, USA) for 30 min at room temperature. Post staining, cells were washed twice and fixed in 4% paraformaldehyde (PFA - Cat. No.: 420801, BioLegend, San Diego, CA, USA) for 30 min at room temperature. Samples were examined and imaged using a Zeiss Axio Observer with an epifluorescence attachment and a digital camera. Five random fields were taken per well and quantified using Image J with colour deconvolution plugin for mean staining area per channel (ver 1.64r).

## Enzyme-linked immunosorbent assay

Sandwich Enzyme-linked immunosorbent assay (ELISA) was used to measure the release of inflammatory cytokines IL-1β, TNFα and IL-18 in the cell supernatant, cell lysate, or mouse plasma post bacterial infection.

For mouse TNFα (Cat. No.: 88-7324-88, Invitrogen) and IL-1β (Cat. No.: 88-7013-88, Invitrogen), 96-well ELISA plates (Cat. No.: 9018, Corning) were prepared per the manufacturer's instructions. For the pre-coated IL-18 ELISA (Cat. No.: BMS618-3, Invitrogen), the experiments were performed according to the manufacturer's instructions.

All plates were measured using TECAN Infinite® 200 Pro (Tecan, Männedorf, Switzerland), with wavelength set at 450 nm.

## RNA extraction and conversion to cDNA

Samples were lysed directly in Trizol reagent (Cat. No.: 15596018, Life Technologies) and stored at −80 °C until RNA extraction. RNA extraction was carried out using Qiagen RNeasy kit (Cat. No.: 74134, Qiagen) according to the manufacturer's instructions. RNA samples were eluted using RNase-free water provided by the kit and then stored at −80 °C. RNA was then converted to cDNA following the manufacturer's instructions for MultiScribe reverse transcriptase (Cat. No.: 4368813, Applied Biosystems). Briefly, RNAse-free water was added to 0.5 µg of total RNA to a final volume of 10 µl. 10 µl of reaction mixture containing random primers, dNTPs (dATP, dGTP, dCTP and dTTP) and MultiScribe reverse transcriptase (Cat. No.: 4368813, Applied Biosystems) were added to the RNA solution. The samples were mixed and heated to 25 °C for 10 min, incubated at 37 °C for 2 h, followed by 85 °C for 5 min in a thermocycler.

## Real-time reverse transcriptase polymerase chain reaction

In all, 10 µl of SSOAdvanced™ Universal SYBR® Green Supermix (Cat. No.: 1725275, Bio-Rad), 0.6 µl of 10 µM forward and reverse primer each (Suppl. Table 1) and nuclease-free water was transferred to each well of a Micro-Amp™ fast optical 96-well reaction plate (Cat. No.: 4346907, Applied Biosystems). Diluted cDNAs were added to wells in duplicate while non-template control wells were also loaded. PCR was performed using ABI StepOne™ real-time PCR system, version 2.1 software program (Applied Biosystems, Foster City, CA, USA). Real-time reverse transcriptase polymerase chain reaction data was analysed using the comparative $2^{-\Delta\Delta CT}$ method[46] with Gapdh as a housekeeping gene.

## Immunoblotting

Post infection, BMDMs and the collected supernatant sample were lysed in Radioimmunoprecipitation assay buffer (RIPA) buffer supplemented with protease inhibitors, i.e., Complete Protease Inhibitor Cocktail Tablets (Cat No.: 04693132001, Roche) to prevent sample degradation. Samples were boiled with 6x Laemmli buffer containing sodium dodecyl sulfate (SDS) and 100 mM dithiothreitol (DTT) for 5 min before storing at −80 °C.

The sample was then thawed on ice and heated to 95 °C for 10 min after thawing. Each sample was loaded on an individual lane of a 4–15% gradient SDS-PAGE gel (Cat No.: 456-1086, Bio-Rad) in SDS running buffer and run

with a constant voltage of 200 volts for ~25 min until the dye front reached the end of the gel. The resolved proteins in the SDS-PAGE gel were then transferred to a 0.45 µm Polyvinylidene fluoride (PVDF) membrane (Cat No.: 1620115, Bio-Rad) by electroblotting. An electric current of 400 mA was applied to the apparatus for 1.5 h at 4 °C. Following the transfer, the membrane was blocked with 5% (w/v) skim milk in PBS for 1 h at room temperature to prevent non-specific binding of Immunoglobulins (Ig).

The PVDF membrane was incubated with primary mouse anti-mouse caspase-1 (1:1000, Cat. No.: 106-42020, Adipogen), caspase-11 (1:1000, Cat. No.: NB120-10454, Novusbio), or Glyceraldehyde 3-phosphate dehydrogenase (GADPH) (1:1000, Cat No.: MAB374, Merck Millipore), GSDMD (1:3000, Cat No.: ab209845, Abcam), diluted in 1% (w/v) skim milk in PBST (PBS with 1% Tween-20) overnight at 4 °C, with gently rocking. PVDF membranes were then incubated with horseradish peroxidase-conjugated secondary antibody (1:5000) for 1 h at room temperature. Immunoreactive proteins were detected by applying ECL Western blotting Detection Reagent (Cat No.: 1705060, Bio-Rad) or SuperSignal™ West Femto Maximum Sensitivity Substrate (Cat. No.: 34096, Thermo Fisher Scientific). The ChemiDoc™Touch Imaging System (BioRad) was used for all blots.

## Recombinant protein expression and purification

The BL21(DE3) *E. coli* strain (C2527H, NEB) was transformed with pET-28a(+)-TEV plasmid containing the sequence for mouse GBP1 (mGBP1) and transformants were selected with 50 µg/ml kanamycin (10106801001, Roche). A single colony was used to inoculate a starter culture of 10 ml $LB_{Kan}$ broth (LB broth + 50 µg/ml kanamycin) which was incubated at 37 °C, shaking (180 rpm) overnight. The overnight culture was diluted 1:100 into 800 ml of $LB_{Kan}$ broth and incubated at 37 °C, shaking (180 rpm) for 2–3 h until an $OD_{600}$ of 0.7 was obtained. Cultures were cooled to room temperature, expression was induced by adding isopropyl β-D-1-thiogalactopyranoside (0.5 mM; IPTG, Roche) and the incubation continued at 18 °C with shaking (180 rpm) overnight. The culture was centrifuged ($5000 \times g$, 20 min, 4 °C) to pellet the bacteria and stored at −80 °C until required. The cell pellet was resuspended in lysis buffer (50 mM $NaH_2PO_4$, 300 mM NaCl, 10 mM imidazole, 5% glycerol (v/v), 5 mM $MgCl_2$, 0.01% Triton X-100, pH 8.0) supplemented with lysozyme (250 µg/ml), Benzonase nuclease (50 U/ml) and protease inhibitor cocktail (11697498001, Roche) and incubated with gentle agitation at 4 °C for 1 h. Cells were subsequently disrupted by sonication and centrifuged ($18,000 \times g$, 30 min, 4 °C) to pellet cellular debris. The supernatant was passed through a 0.22 µm filter (SLGP033RS, Merck) and mGBP1 was purified using Ni-NTA agarose resin (30210, Qiagen) as per the manufacturers' instructions. The purity of eluted proteins was analysed by SDS-PAGE and Coomassie blue staining. Purified proteins were dialysed in DPBS (14190, ThermoFisher) containing 20 mM Tris and 20% glycerol (v/v), pH 7.5.

## Antimicrobial assays

For bacterial viability assays, overnight cultures of *A. baumannii* were washed and resuspended with PBS to a concentration of $1 \times 10^6$ CFU/ml. Bacteria were then treated with solvent control (PBS), recombinant GBP1 (10-320 µg/ml) or the positive control peptide WLBU2 (25 µg/mL) and incubated at 37 °C for 6 h. Treated bacteria were serially diluted, plated onto trypticase soy agar plates, and incubated overnight at 37 °C. Colonies were enumerated the following day.

## Mice

C57BL/6 mice and $Gsdmd^{-/-}$ mice carrying a missense mutation impairing pore formation but not proteolytic cleavage[11] were sourced from The Australian National University. $Nlrp3^{-/-47}$, $Casp1^{-/-48}$, $Casp1/11^{-/48}$, $Casp11^{-/-49}$ and $Ifnar^{-/-50}$ mice were sourced from The Jackson Laboratory. $Asc^{-/-9}$ mice were sourced from the University of Queensland. $Gbp1^{-/-}$ mice were generated by CRISPR-Cas9 gene editing technology and was previously described[21].

**Article**

All mice are on, or backcrossed to, the C57BL/6 background for at least 10 generations. Mice of 8-12-weeks old were used. Mice were bred and maintained at The Australian National University under specific pathogen-free conditions. We have complied with all relevant ethical regulations for animal use. All animal studies were performed in accordance with the National Health and Medical Research Council code for the care and use of animals under the Protocol Number A2018-08 and A2021-14 approved by The Australian National University Animal Experimentation Ethics Committee.

### In vivo infection

*A. baumannii* was streaked onto trypticase soy agar plates and incubated at 37 °C overnight for isolation of single colony. Single colonies of *A. baumannii* were picked and inoculated into 5 ml of trypticase soy broth and incubated at 37 °C 16-18 h on shaker at 220 rpm for bacterial propagation. 5 ml of the bacterial broth was diluted 1:5 with fresh trypticase soy broth the next day and incubated for further 2 h to ensure most of the bacteria cells are in log phase of growth. After incubation, bacteria were collected and washed with sterile 1xPBS at 2,800 rcf for 30 min. Mice were infected via intraperitoneal injection of *A. baumannii* (200 µl/mouse, $2 \times 10^7$ CFU/mouse). The mice were monitored every 4 h until 28 h post infection. Observations consistent with illness during monitoring include coat condition (ruffles), hydration levels (whether the mice were eating or drinking) and activity level (whether the mice are moving, i.e. if there's slowing in movement). Each of these categories was scored independently between 0 (normal) and 2 (very ill).

Mice were humanely euthanised when they were considered a score of 2 for any of these categories. Approximately 20–50 mg of organs (20–50 mg of liver, one lung, the spleen and one kidney) were isolated, weighed and filtered through a 70 µm nylon mesh cell strainer in 1 ml of sterile 1xPBS before serial dilutions in 1xPBS and enumeration on trypticase soy agar plates.

### Characterisation of effector cell populations in mice lungs post A. baumannii infection

One lung was excised from an infected mouse and rinsed with sterile 1xPBS. The lung was then finely diced and placed in 1 ml of 1 mg/ml Collagenase P/RPMI (Cat. No.: 11-213-873-001, Roche) and incubated at 37 °C for 30 min. The digested lung was mashed using a 3 ml syringe plunger through a 70-µm nylon mesh cell strainer. The resultant cell suspension was centrifuged at 300 rcf for 5 min. The cell pellet was resuspended in 0.5 ml of 1x RBC lysis buffer and centrifuge at 300 rcf for 5 min. Lung cells were transferred to a 1.5 ml microfuge tube for cell surface marker staining.

For intracellular viability staining, 100 µl of Zombie Aqua (1:1000, Cat. No.: 423101, BioLegend, San Diego, CA, USA) was added to each sample for 10 min at room temperature. Cells were washed with 1xPBS before proceeding with cell surface marker staining. The cells were treated with Fc block (Cat. No.: 553142, BD Pharmingen) (5 µl/sample) for 10 min on ice. After incubation, Fc block was removed, and a cocktail of conjugated primary antibodies Ly6g-APC-Cy7 (1:500, Cat. No: 127624, BioLegend), CD11b-PE-Texas Red (1:500, Cat. No: 101256, Biolegend), Ly6c-Alexa Fluor 405 (1:500, Cat. No: 48-5932-82, Thermo), was added directly into relevant samples for 30 min on ice in the dark. The cells were washed with MTRC buffer and centrifuged for 5 min, prior to fixation with 4% PFA for 30 min at room temperature in the dark. The total cell population was collected on FACS Fortessa platform (Becton Dickinson). The analysis was performed using FlowJo software (ver. 10.8.1). The cell population was gated using forward scatter and side scatter to exclude debris, followed by doublet exclusion to characterise single cells (Supplementary Fig. 1a). This population was then gated for different neutrophil populations (Supplementary Fig. 1c, d) and inflammatory monocytes (Supplementary Fig. 1b, d) according to the cell surface marker expression. When characterising cell death using Zombie aqua, cells were further gated based on Zombie aqua fluorescence (Supplementary Fig. 1b, c).

### Statistics and reproducibility

All results provided were performed from at least two independent experiments (biological replicates). The GraphPad Prism 8.0 software was used for data analyses. Data are shown as the mean ± SEM. Statistical significance was determined by t-tests (two-tailed) for two groups or one-way ANOVA (with Dunnett's or Tukey's multiple comparisons tests) for three or more independent groups. Survival curves were compared using the log-rank test. A $p$-value $< 0.05$ was considered statistically significant.

### Reporting summary

Further information on research design is available in the Nature Portfolio Reporting Summary linked to this article.

### Data availability

The data that support the findings of this study are available in the methods and/or supplementary material of this article. All the source data are supplied with the paper Numerical sources of all graphs are provided in Supplementary Data 1. Uncropped and unedited gel images supporting the Figs. 1c and 7a are available in respectively Supplementary Figs. 9 and 10.

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

## Acknowledgements
The authors would like to thank Prof John Boyce and Mrs Amy Wright (Monash University, Australia) for providing the strains 19606, 19606R, AL 1847 and 19606+LpxA. The authors thanks Dr Harpreet Vora, Mr Michael Devoy from the flow cytometry at JCSMR and the Australian Phenomics facility for technical assistance. F.-J.L. was supported by an ANU-Taiwan scholarship.

## Author contributions
F.-J.L., L.S. and G.B. conceived the study. F.-J.L., L.S., A.M. and D.E.T. performed the experiments. F.-J.L., L.S., S.M.M. and G.B. conducted the analysis. S.M.M. provided mice and cells. F.-J.L. and G.B. wrote the manuscript. G.B. provided the overall supervision of the work. All authors provided feedback and approved the manuscript.

## Competing interests
S.M.M. is an Editorial Board Member for *Communications Biology* but was not involved in the editorial review of, nor the decision to publish, this article. The other authors declare no competing interests.
