## [Transparent Peer Review file · Communications Biology]

Interferon signalling and non-canonical inflammasome activation promote host protection against multidrug-resistant *Acinetobacter baumannii*

Corresponding Author: Dr Gaetan Burgio

Version 0:

Reviewer comments:

Reviewer #1

(Remarks to the Author)

In the present study, the authors have demonstrated that type I IFN signaling and non-canonical inflammasome confer host protection against *Acinetobacter baumannii* infection. It appears that the experiments are well-designed and the results are clear. However, some results differ from previous studies and there are several points to be revised.

1. In supplementary Figure 3, only the level of IL-18, not IL-1 β and TNF- α , was significantly lower in GSDMD-deficient BMDMs compared to WT cells. I wonder whether GSDMG specifically regulates IL-18 expression among cytokines.
2. In Figure 2 and supplementary Figure 2, a single deficiency of either caspase-1 or caspase-11 protected mice from lethality by *A. baumannii* infection, while a double deficiency did not offer this protection. The reasons or possible mechanisms behind these results should be discussed. In lines 244-246, the authors suggested that reduced programmed cell death in immune cells may have contributed to decreased host mortality. However, neutrophil cell death was lower in caspase-1/-11 DKO mice, as shown in Figure 4a, and the average percentage of cell death in inflammatory monocytes was also lower in DKO mice compared to WT mice, although not significantly so.
3. In lines 210-212, the sentence should be double-checked, as the percentage of CD11b+Ly6g+ cells is significantly lower in caspase-1/-11 DKO mice, as shown in Figure 3a.
4. In Figure 5e, the bacterial burden in the lungs was significantly reduced in IFNAR-deficient mice. In contrast, the study by Li et al. (Ref 28) reported an increase in bacterial CFU in IFNAR-deficient mice compared to WT mice, which appears to be attributable to reduced inflammation. What can be a reason for these differences?
5. In lines 301-302, is the subtitle correct? I believe that deficiency in GBP1 confers protection against *A. baumannii* infection in mice.

Reviewer #2

(Remarks to the Author)

In this study, Li et al demonstrated that a systemic MDR *A. baumannii* infection and found that MDR *A. baumannii* activates the NLRP3 inflammasome complex predominantly via the non-canonical caspase-11-dependent pathway. They also found that caspase-1 and caspase-11-deficient mice are protected from a virulent MDR *A. baumannii* strain by maintaining a balance between protective and deleterious inflammation. In addition, they found that cytosolic immunity mediated by guanylate-binding protein 1 (GBP1) and type I interferon signaling regulates caspase-11-dependent inflammasome activation.

In summary, they found that Caspase-1/11, ASC, IFN-I, and GBP1 played negative roles in host defense against MDR *A. baumannii* strain infection.

Overall, this study provided novel insight into the appropriate inflammasome activation that was critical for host defense. Excess inflammasome activation was previously reported to be detrimental to host defense against other pathogens, such as *Francisella novicida* mutant strain.

However, type I interferon and inflammasome were previously demonstrated to play positive roles in host defense against *A. baumannii* infection. Why do the inflammasome and IFN-I instead play negative roles against this MDR *A. baumannii* strain infection? Whether this MDR *A. baumannii* strain trigger much higher inflammasome activation or become intracellular bacteria compared with other *A. baumannii* strains?

In addition, the cell population labeled with Ly6c and Cd11b in Figure 3b and Figure 4b needs to be introduced in more detail. Why this population was analyzed, which is not a typical innate immune cell population?

Reviewer #3

(Remarks to the Author)

Acinetobacter baumannii infections are a major concern due to the emergence of multi-drug resistant strains. Here, the authors investigated the role of innate immunity in the response to *A. baumannii* infection in mice. The authors found that in BMDMs, *A. baumannii* causes an activation of the non-canonical inflammasome pathway (e.g. Caspase-11-NLRP3-Caspase-1). They next examined the response of mice deficient in the components of this signaling pathway and found that deficiency in Casp11 or Casp1 protected mice from *A. baumannii*-induced death, while NLRP3, ASC or GSDMD KO animals showed a similar trend. They also found lower CFU, mainly in the Casp11 KO mice and low production of IL-18 and IL-1b. Intriguingly though, double deficiency in both Casp1 and Casp11 (Casp1/11 dKO) did not protect the mice from death, while reducing cytokines. To understand the phenotype of Casp1/11 dKO animals, the authors investigated what underlies resistance to *A. baumannii*, and propose that inflammasome-driven death of monocytes and neutrophils reduces the ability of mice to resist the infection. Finally, the authors show that type-I-IFN signaling and the IFN-induced GTPase mGBP1 promote non-canonical inflammasome activation during *A. baumannii* infections. In summary, the data support the conclusion that activation of the non-canonical inflammasome during *Acinetobacter baumannii* infections is detrimental to the host, which confirms previous data showing that deficiency in NLRP3 inflammasome components protects against lung pathology after *A. baumannii* infections (Kang et al. Immunology 2017).

Overall, this is a mostly well-done study that dissect the contribution of different inflammasome proteins to host defense against *A. baumannii* infections. It expands on previous work that has shown a role for the NLRP3 inflammasome and caspase-11 (Kang et al. Immunology 2017, Li Microb Path 2021, Wang Inf Imm 2017), and supports the Kang et al. that reported a detrimental role for inflammasome activation. In addition, the paper shows that IFN α signaling and mGBP1 are important for Caspase-11 activation, thus going beyond previous work. My major criticism of the study relates to 1) the mechanism by which the non-canonical pathway is detrimental to mice, 2) the curious phenotype of the Casp1/-11 dKO mice, the role of mGBP1 and 4) the quality of the in vitro assays. In addition, the data seems to some degree overinterpreted and some of the explanation do frankly not make sense to me. Below I provide more detail on this criticism. Improving these points will be necessary before I support publication.

Major points:

In vitro assays: While the in vivo data looks solid and show the unavoidable level of mouse-to-mouse variability, the in vitro assays (immunoblots, ELISAs, etc..) should show more consistency and several of them need improvement:

Fig. 1a: Why are TNF levels reduced in Casp11 KO cells by 50%? TNF levels are highly variable. This is unusual and maybe due to a seeding error? (of note, we have seen that Casp11 KO BMDMs grow more slowly than WT, maybe a normalization for cell numbers is necessary?).

Fig. 1c: The Casp1 blot looks good, but Casp11 and GSDMD blots are unconvincing. Why is there Casp11 p26 in uninfected cells? Same for GSDMD p30. Actually I can see that uninfected cells show both GSDMD-FL and p30 and that both bands get more intense after infection. (consider using another GSDMD Ab).

Fig. 1d: The cell death data with the Gsdmd KO are not convincing. Since GSDMD will eventually undergo death due to apoptosis activation, I would suggest to redo the assay using Casp1/11 dKO BMDMs.

Fig. 7a: why is there no p20 in these blots? If there is Casp11 activation and GSDMD cleavage – wouldn't Caspase-1 get activated as well?

Fig. 7a: The GBP1 KO BMDMs seem to have significantly lower expression of GSDMD, even though more lysate was loaded (based on GAPDH levels). This is highly troubling as such a reduction of GSDMD expression could account for the observed phenotype. Please verify that GSDMD expression is similar between WT and GBP1 KOs.

Fig 7b: Similarly, pro-Caspase-11 levels are lower in GBP1 KOs after infection (see both Casp11 FL bands). Why is that the case? Does deficiency in mGBP1 result in lower IFN production? Can the authors confirm that their CRISPR deletion didn't result in off-target effects?

Fig 1 and 7: Please show the LDH data as normally done in the field as percentage of full lysis controls and not OD values. It is unclear from the OD values how much cell death could be observed.

Phenotype of Casp1/11 dKO animals: The finding that Casp1/11 dKO animals are comparable to WT animals in regard to survival is curious. The authors propose that the finding that Casp1/11 double deficiency does not confer protection could be explained by a lack of a protective inflammation (low levels of IL1b/18). But the Casp11 KO mice have similarly low levels of IL-1b/18 and nevertheless they are protected. Thus, this explanation alone falls short, and I would even argue that inflammation is what primarily kills the mice, since Casp1 KOs are similarly protected from death as Casp11 KOs.

Suppl. Fig 2d: The authors also highlight that low level of CCR2 in Casp1/11 mice correlates with low inflammation in these mice. But NLRP3 KOs show even lower levels of CCR2 than Casp1/11 and still some inflammation. Thus, there can't be any relationship between CCR2 and inflammation levels.

It is possible that I did not understand the authors model for the observed phenotypes of the different mice (maybe including a schematic would help the readers). To me it seems that, if we consider that casp1 sKO are protected and that Casp1 is not activated in Casp11 KOs, then Casp1-driven cytokine over-production must be the main cause of death in mice. However, pyroptosis is still important as it keeps the bacterial numbers in check. This pyroptosis is mainly driven by Casp-11 (Casp-11 drives pyroptosis via GSDMD, and in a separate pathway NLRP3-ASC-Caspase-1 driven IL-1b/18 production (see Kayagaki et al. 2011)), but the NLRP3-ASC-Casp1 inflammasome contributes to some degree as CFUs are also reduced in absence of N3 and ASC. This goes along with the authors model that maintaining a balance between protective and deleterious inflammation is important.

But why are the Casp1/11 dKO no longer protected, and what activates casp-1 in the Casp11 KO mice so that Casp1 can provide a beneficial and protective inflammation? My best explanation is that there is in vivo (but not in BMDMs) a separate canonical inflammasome pathway activated (maybe NLRC4, since ASC KOs are similar to Nlrp3 KOs) that provides low level cytokine production in absence of Caspase-11 and thus protection.

Mechanistic explanation for how inflammasome activation proves to be detrimental to mice: I think that the CFU changes in inflammasome-deficient mice are indeed best explained by a loss of immune cells through pyroptosis. But I am not convinced that the data shown in Fig 4a/b really allow this conclusion. The authors conclude that in NLRP3, ASC and Casp11 lower level of dying neutrophils and monocytes could cause the reduction in bacterial burden. However, this conclusion is not convincing as: 1) NLRP3 KO mice do not have reduced monocyte levels. Actually they have even more than WT, which is left unexplained, 2) Casp1/11 dKO show similarly reduced levels of neutrophils and partially reduced levels of monocytes, and still do have high levels of CFUs.

Maybe additional experiments that would more convincingly show a loss of a certain cell type due to pyroptosis could help. Or alternatively, such a loss could be introduced artificially (deplete neutrophils for example) and show that this is detrimental during infection.

Role of GBPs:

a) The phenotype of the GBP1 KO mice mimics Casp11 deficiency, but the title of that chapter is confusing as it claims that mGBP1 protects the host – wouldn't it be better to say that it is detrimental?

b) Also, I do not understand how the authors can conclude that mGBP1 does not protect via direct killing? I don't think that this is the correct conclusion from the data presented. The data show that changes in the LOS (or complete loss of LOS) do not reduce Casp11 activation. Since LPS is the only known ligand of both GBPs and caspase-11 (-4), the correct conclusions must be: 1) that mGBP1 does not recognize LOS, but some other signal stemming from A.b. infections, and 2) that Casp11 does not recognize LOS but some other signal stemming from A.b. infections. Such a conclusion is quite unexpected and would be highly controversial, and would need of course additional data to convince the field.

Minor points:

Fig 5d/e: The description of the data remain misleading,, as the authors do not point out that IFNAR deficiency has no impact on CFU in other organs save the lung.

Version 1:

Reviewer comments:

Reviewer #1

(Remarks to the Author)

The authors well addressed all issues raised by the reviewer.

Reviewer #2

(Remarks to the Author)

My concerns have been addressed.

Reviewer #3

(Remarks to the Author)

The authors present a much improved version of their manuscript, where they addressed the points raised by the referees. I have no additional comments.

Referee expertise:

Referee #1: *Acinetobacter baumannii*; nlrp3 inflammasome; inflammation

Referee #2: nlrp3; *Acinetobacter baumannii*; immunity

Referee #3: inflammasome; pyroptosis; innate immunity

Reviewers' comments:

Reviewer #1 (Remarks to the Author):

In the present study, the authors have demonstrated that type I IFN signaling and non-canonical inflammasome confer host protection against *Acinetobacter baumannii* infection. It appears that the experiments are well-designed and the results are clear. However, some results differ from previous studies and there are several points to be revised.

1. In supplementary Figure 3, only the level of IL-18, not IL-1 β and TNF- α , was significantly lower in GSDMD-deficient BMDMs compared to WT cells. I wonder whether GSDMD specifically regulates IL-18 expression among cytokines.

Response: We thank the reviewer for this comment and taking the time in reviewing the manuscript. Indeed, a recent paper has demonstrated that GSDMD specifically regulate IL-18 to control its release via Caspase 4 in human cells. <https://www.biorxiv.org/content/10.1101/2024.02.01.578487v1.full>. We added a mention of this finding in our results section L177 "While the bacterial burden in the *Gsdmd*^{-/-} mice did not differ from WT mice, we observed a 10-fold reduction in plasmatic IL-18 in *Gsdmd*^{-/-} mice (Supp. Fig. 3b) in agreement with a recent report²⁶"

2. In Figure 2 and supplementary Figure 2, a single deficiency of either caspase-1 or caspase-11 protected mice from lethality by *A. baumannii* infection, while a double deficiency did not offer this protection. The reasons or possible mechanisms behind these results should be discussed. In lines 244-246, the authors suggested that reduced programmed cell death in immune cells may have contributed to decreased host mortality. However, neutrophil cell death was lower in caspase-1/-11 DKO mice, as shown in Figure 4a, and the average percentage of cell death in inflammatory monocytes was also lower in DKO mice compared to WT mice, although not significantly so.

Response: We do agree with the reviewers the results on caspase-1/-11 DKO are counterintuitive and unexpected. We tried to rationalise it by quantifying the cell death in neutrophils/monocytes populations in the lungs of infected mice. As the referee mentioned, we found a reduced cell death in caspase -1/-11 DKO. However, while reduced, the magnitude was intermediate between WT and single KO mice for monocytes and more pronounced for neutrophils. However, the recruitment of the neutrophils/monocytes is significantly different between the single KO (more recruitment) versus the DKO (less recruitment). Taken together, the balance recruitment versus cell death indicates the presence of less effectors in caspase -1/-11 DKO versus the single KO. To capture the referee's comment and clarify our findings we amended the manuscript as followed: L213-215 "we observed no significant difference in the percentage of CD11b⁺Ly6g⁺ and CD11b⁺Ly6c⁺ in the lungs between WT, *Asc*^{-/-}, *Casp1/11*^{-/-} and *Casp11*^{-/-} mice but noticed a reduction of cell numbers in *Casp1/11*^{-/-} mice (Fig. 3a and 3b)". L224-225 "All knock-out mouse strains showed a significant decrease in the number of dead or dying neutrophils including *Casp1/11*^{-/-} mice (Fig 4a)" and L262-264 "These data collectively suggest that in *Nlrp3*^{-/-}, *Asc*^{-/-}, *Casp11*^{-/-} and *Casp1/11*^{-/-} mice, the balance between effector recruitment (increase cell recruitment) and programmed cell death (lower cell death) might have contributed to lower bacterial burden and decreased host mortality."

3. In lines 210-212, the sentence should be double-checked, as the percentage of CD11b+Ly6g+ cells is significantly lower in caspase-1/-11 DKO mice, as shown in Figure 3a.

Response: We thank the reviewer for spotting the issues. We fixed it in L213 (please see previous comment)

4. In Figure 5e, the bacterial burden in the lungs was significantly reduced in IFNAR-deficient mice. In contrast, the study by Li et al. (Ref 28) reported an increase in bacterial CFU in IFNAR-deficient mice compared to WT mice, which appears to be attributable to reduced inflammation. What can be a reason for these differences?

Response: We thank the referee for this interesting point. The work from Li et al. uses a different strain, a clinical *A. baumannii* strain, different MOI, different mode of injection (intranasal) and different timepoint (8 hours post inoculation). Our results indeed are either in agreement or contrasting with recently published works that have

demonstrated that type I IFN response could be either beneficial (Akoolo et al. J innate immunity 2022 14:544-553) or detrimental (Li et al. Cell Death and Differentiation 2018 25:1304-18). This discrepancy could be due to differences in experimental settings. As we observed differences were consistent across our experimental settings, we ruled out this hypothesis. As mentioned from the referee, this discrepancy in outcome from Li et al. is likely due to the strain virulence. We indeed noticed a variability in the inflammation outcome in our previous publication (Li et al. Plos One 2022 17:e0277019). Our explanation is the differential outcome is likely link to the outer membrane, vesicle and capsule composition/virulence. Having noticed these differences, we were cautious in our description of the inflammatory response not to generalise to all *A.baumannii* isolates.

We added the following section in the discussion section L440-446 "The virulence of the bacterial strain outer membrane, vesicle/capsule composition, the mode of infection and the bacterial concentration differ from this previous report, which has resulted in a differently observed outcome. Our findings and our previous assessment on multiple strains with different growth rate and inflammatory responses³⁸ however, clearly establishes that the non-canonical inflammasome activation via caspase-11-mediated cell death are major host defence mechanisms against MDR *A. baumannii*"

5. In lines 301-302, is the subtitle correct? I believe that deficiency in GBP1 confers protection against *A. baumannii* infection in mice.

Response: We corrected the subtitle by adding "Deficiency in IFN-inducible..."

Reviewer #2 (Remarks to the Author):

In this study, Li et al demonstrated that a systemic MDR *A. baumannii* infection and found that MDR *A. baumannii* activates the NLRP3 inflammasome complex predominantly via the non-canonical caspase-11-dependent pathway. They also found that caspase-1 and caspase-11-deficient mice are protected from a virulent MDR *A. baumannii* strain by maintaining a balance between protective and deleterious inflammation. In addition, they found that cytosolic immunity mediated by guanylate-binding protein 1 (GBP1) and type I interferon signaling regulates caspase-11-dependent inflammasome activation.

In summary, they found that Caspase-1/11, ASC, IFN-I, and GBP1 played negative roles in host defense against MDR *A. baumannii* strain infection.

Overall, this study provided novel insight into the appropriate inflammasome activation that was critical for host defense. Excess inflammasome activation was previously reported to be detrimental to host defense against other pathogens, such as *Francisella novicida* mutant strain.

However, type I interferon and inflammasome were previously demonstrated to play positive roles in host defense against *A. baumannii* infection. Why do the inflammasome and IFN-I instead play negative roles against this MDR *A. baumannii* strain infection? Whether this MDR *A. baumannii* strain trigger much higher inflammasome activation or become intracellular bacteria compared with other *A. baumannii* strains?

Response: We thank the referee for this comment and taking the time to review the manuscript. Our results indeed are contrasting with recently published works that have demonstrated that type I IFN response could be either beneficial (Akoolo et al. J innate immunity 2022 14:544-553) or detrimental (Yi et al. Cell Death and Differentiation 2018 25:1304-18). This discrepancy could be due to differences in experimental settings. We double checked again by infecting males and females and the differences (total number of mice 42) across multiple strains and the observed differences were consistent across our experimental setting. As mentioned from the referee, this discrepancy in outcome is likely due to the strain virulence. We indeed noticed a variability in the inflammation outcome across strains in our previous publication (Li et al. Plos One 2022 17:e0277019). The strain 1605 indeed display a higher level of inflammation from ATCC 17978 and 19606 strains (Li et al Plos One Fig 3) and highest growth rate (Li et al. Plos One FigS3). It could explain indeed differences in IFN response.

We modified our discussion section to reflect this point L440-446 "The virulence of the bacterial strain outer membrane, vesicle/capsule composition, the mode of infection and the bacterial concentration differ from this previous report, which has resulted in a differently observed outcome. Our findings and our previous assessment on multiple strains with different growth rate and inflammatory responses³⁸ however, clearly establishes that the non-canonical inflammasome activation via caspase-11-mediated cell death are major host defence mechanisms against MDR *A. baumannii*"

In addition, the cell population labeled with Ly6c and Cd11b in Figure 3b and Figure 4b needs to be introduced in more detail. Why this population was analyzed, which is not a typical innate immune cell population?

Response: The Ly6c, Ly6g and Cd11b population have been previously demonstrated as important first line of effectors against bacterial population in response to acute bacterial infection and sepsis in the tissues. These cells emerge from the periphery to produce cytokines (type I interferon) and to exert their phagocytic functions. It was demonstrated in previous work that these cells are required for bacterial clearance of *Acinetobacter baumannii* infection (Pires et al. JCI insights 2020 5(7):e132223). We added the following in L215-218. "We

postulated that first line of cellular effectors against acute bacterial infection and sepsis to produce type I interferon cytokines and exert their phagocytic functions are neutrophils (CD11b⁺Ly6g⁺) and inflammatory monocytes (CD11b⁺Ly6c⁺)²⁸. We hypothesise these immune cell effectors are required for bacterial clearance by increase recruitment and/or increased clearance of infected cells in the target tissues, such as the lung, were responsible for enhanced resistance to *A. baumannii* 1605”

Reviewer #3 (Remarks to the Author):

Acinetobacter baumannii infections are a major concern due to the emergence of multi-drug resistant strains. Here, the authors investigated the role of innate immunity in the response to *A. baumannii* infection in mice. The authors found that in BMDMs, *A. baumannii* causes an activation of the non-canonical inflammasome pathway (e.g. Caspase-11-NLRP3-Caspase-1). They next examined the response of mice deficient in the components of this signaling pathway and found that deficiency in Casp11 or Casp1 protected mice from *A. baumannii*-induced death, while NLRP3, ASC or GSDMD KO animals showed a similar trend. They also found lower CFU, mainly in the Casp11 KO mice and low production of IL-18 and IL-1 β . Intriguingly though, double deficiency in both Casp1 and Casp11 (Casp1/11 dKO) did not protect the mice from death, while reducing cytokines. To understand the phenotype of Casp1/11 dKO animals, the authors investigated what underlies resistance to *A. baumannii*, and propose that inflammasome-driven death of monocytes and neutrophils reduces the ability of mice to resist the infection. Finally, the authors show that type-I-IFN signaling and the IFN-induced GTPase mGBP1 promote non-canonical inflammasome activation during *A. baumannii* infections. In summary, the data support the conclusion that activation of the non-canonical inflammasome during *Acinetobacter baumannii* infections is detrimental to the host, which confirms previous data showing that deficiency in NLRP3 inflammasome components protects against lung pathology after *A. baumannii* infections (Kang et al. Immunology 2017).

Overall, this is a mostly well-done study that dissect the contribution of different inflammasome proteins to host defense against *A. baumannii* infections. It expands on previous work that has shown a role for the NLRP3 inflammasome and caspase-11 (Kang et al. Immunology 2017, Li Microb Path 2021, Wang Inf Imm 2017), and supports the Kang et al. that reported a detrimental role for inflammasome activation. In addition, the paper shows that IFN α signaling and mGBP1 are important for Caspase-11 activation, thus going beyond previous work. My major criticism of the study relates to 1) the mechanism by which the non-canonical pathway is detrimental to mice, 2) the curious phenotype of the Casp1/11 dKO mice, the role of mGBP1 and 4) the quality of the in vitro assays. In addition, the data seems to some degree overinterpreted and some of the explanation do frankly not make sense to me. Below I provide more detail on this criticism. Improving these points will be necessary before I support publication.

Major points:

In vitro assays: While the in vivo data looks solid and show the unavoidable level of mouse-to-mouse variability, the in vitro assays (immunoblots, ELISAs, etc..) should show more consistency and several of them need improvement:

Fig. 1a: Why are TNF levels reduced in Casp11 KO cells by 50%? TNF levels are highly variable. This is unusual and maybe due to a seeding error? (of note, we have seen that Casp11 KO BMDMs grow more slowly than WT, maybe a normalization for cell numbers is necessary?).

Response: We thank the referee of taking the time to carefully review the manuscript. The data is indeed across different rounds of experiment causing indeed a seeding error for the ELISA assay. While we don't have a clear idea why TNF level was reduced in Casp11 KO cells, it suggests an overall reduction in the inflammation level and perhaps less of a feedback loop from these inflammatory mediators

Fig. 1c: The Casp1 blot looks good, but Casp11 and GSDMD blots are unconvincing. Why is there Casp11 p26 in uninfected cells? Same for GSDMD p30. Actually I can see that uninfected cells show both GSDMD-FL and p30 and that both bands get more intense after infection. (consider using another GSDMD Ab).

Fig. 1d: The cell death data with the Gsdmd KO are not convincing. Since GSDMD will eventually undergo death due to apoptosis activation, I would suggest to redo the assay using Casp1/11 dKO BMDMs.

Response: Due to the overnight infection conditions, minor autoproteolytic cleavage of Caspase-11 and GSDMD in the unstimulated controls may occur due to prolonged stress in unstimulated macrophages. We have now provided unstimulated control ELISA data to supplement Figure 1a to show that these minor cleavage events do not contribute to cytokine production.

Fig. 7a: why is there no p20 in these blots? If there is Casp11 activation and GSDMD cleavage – wouldn't Caspase-1 get activated as well?

Response: Despite several attempts for this revision, we failed to optimise the p20 blots in Figure 7a. We have removed this blot along with the associated text mentioning caspase-1 cleavage for Figure 7a.

Fig. 7a: The GBP1 KO BMDMs seem to have significantly lower expression of GSDMD, even though more lysate was loaded (based on GAPDH levels). This is highly troubling as such a reduction of GSDMD expression could account for the observed phenotype. Please verify that GSDMD expression is similar between WT and GBP1 KOs.

Response: We have previously confirmed that GSDMD expression between WT BMDMs and GBP1-KO BMDMs is similar (Enosi Tuipulotu et al EMBOJ, 2023; Figure 5, Source Data).

Fig 7b: Similarly, pro-Caspase-11 levels are lower in GBP1 KOs after infection (see both Casp11 FL bands). Why is that the case? Does deficiency in mGBP1 result in lower IFN production? Can the authors confirm that their CRISPR deletion didn't result in off-target effects?

Response: We have previously confirmed that caspase-11 expression between WT BMDMs and GBP1-KO BMDMs is similar (Enosi Tuipulotu et al EMBOJ, 2023; Figure 5A). CRISPR deletion of target effects was investigated, verified by sequencing (Feng et al. Nature Communications 2022; 13:4395 Supplementary Tables 1 and 2) confirming that no off-target effects have been observed on GBP1-KO CRISPR edited mice.

Fig 1 and 7: Please show the LDH data as normally done in the field as percentage of full lysis controls and not OD values. It is unclear from the OD values how much cell death could be observed.

Response: We have been unable to make this change. Therefore, we have removed the LDH measurement in Figures 1 and 7 and mention to LDH measurement in the manuscript.

Phenotype of Casp1/11 dKO animals: The finding that Casp1/11 dKO animals are comparable to WT animals in regard to survival is curious. The authors propose that the finding that Casp1/11 double deficiency does not confer protection could be explained by a lack of a protective inflammation (low levels of IL1b/18). But the Casp11 KO mice have similarly low levels of IL-1b/18 and nevertheless they are protected. Thus, this explanation alone falls short, and I would even argue that inflammation is what primarily kills the mice, since Casp1 KOs are similarly protected from death as Casp11 KOs.

Response: We thank the referee for this comment. We are grateful to the reviewer to help us in better understand why the Casp1/11 dKO survived to the infection. While indeed IL-1b level is similar between Casp11 and Casp1/11, detailed examination evidenced that IL-18 is abolished in Casp1/11 dKO whereas reduced in Casp1 and Casp11 KO with similar TNF levels. Therefore, the referee's interpretation is the most likely explanation.

We made changes in the following sections. L180 "Additionally, these data confirmed that while NLRP3/ASC, caspase-1 or caspase-11 deficiency alone conferred mouse survival whereas deficiency in both caspase-1 and caspase-11 did not, possibly due to the overall reduction of IL-1β or IL-18 and increase in TNFα secretion in caspase 1/11 deficiency" and L420 "While IL-1β and IL-18 pro-inflammatory cytokines secretion were reduced or

abolished in caspase knockout mice, other pro-inflammatory inflammasome independent cytokines secretion (TNF α and IL-6) were maintained or elevated. It suggests that a balance between a lower to abolished inflammasome dependent pro-inflammatory cytokine secretion and elevated TNF α and IL-6 secretion is likely causative for caspase 1/11 double knockout death."

Suppl. Fig 2d: The authors also highlight that low level of CCR2 in Casp1/11 mice correlates with low inflammation in these mice. But NLRP3 Kos show even lower levels of CCR2 than Casp1/11 and still some inflammation. Thus, there can't be any relationship between CCR2 and inflammation levels.

Response: In Figure 3d, NLRP3 KO shows a similar level to WT whereas Casp1/11 dKO is significantly lower than other strains. In suppl Figure 7, CCR2 level is shown for only IFNAR KO mice and not Caspase 1/11 double knockout. We therefore stand by our conclusion that Caspase 1/11 display low level of inflammation.

It is possible that I did not understand the authors model for the observed phenotypes of the different mice (maybe including a schematic would help the readers). To me it seems that, if we consider that casp1 sKO are protected and that Casp1 is not activated in Casp11 KOs, then Casp1-driven cytokine over-production must be the main cause of death in mice. However, pyroptosis is still important as it keeps the bacterial numbers in check. This pyroptosis is mainly driven by Casp-11 (Casp-11 drives pyroptosis via GSDMD, and in a separate pathway NLRP3-ASC-Caspase-1 driven IL-1b/18 production (see Kayagaki et al. 2011)), but the NLRP3-ASC-Casp1 inflammasome contributes to some degree as CFUs are also reduced in absence of N3 and ASC. This goes along with the authors model that maintaining a balance between protective and deleterious inflammation is important.

Response: We thank the referee for this point. The model proposed from the referee is in line to our model. To clarify our mode to the readership we followed the referee's recommendation and added a schematic of the model (Figure 8)

But why are the Casp1/11 dKO no longer protected, and what activates casp-1 in the Casp11 KO mice so that Casp1 can provide a beneficial and protective inflammation? My best explanation is that there is in vivo (but not in BMDMs) a separate canonical inflammasome pathway activated (maybe NLRC4, since ASC KOs are similar to Nlrp3 KOs) that provides low level cytokine production in absence of Caspase-11 and thus protection.

Response: We thank the referee for this point. We have indeed thought on a distinct pathway explaining why Casp1/11 KO is no longer protected and we indeed hypothesise that a second pathway is implicated. We tested multiple pathways in vitro (Figure 1). For in vivo infections, previous work demonstrated that NLRC4 KO and WT were similar throughout the infection (Kang et al. Immunology 2017: 495-505 – Figure 1). We have performed preliminary experiments on other pathways during the revision of the manuscript (pyrin, cGAS, Aim2...) with a few mice per strains and to date we couldn't observe any differences to WT either (not able to add to the manuscript due to the low sample size). While we agree with the referee and suspected as well that an additional pathway might play an important role in the response to A.baumannii, we unfortunately haven't been able to substantiate this claim.

Mechanistic explanation for how inflammasome activation proves to be detrimental to mice: I think that the CFU changes in inflammasome-deficient mice are indeed best explained by a loss of immune cells through pyroptosis. But I am not convinced that the data shown in Fig 4a/b really allow this conclusion. The authors conclude that in NLRP3, ASC and Casp11 lower level of dying neutrophils and monocytes could cause the reduction in bacterial burden. However, this conclusion is not convincing as: 1) NLRP3 KOs do not have reduced monocyte levels.

Actually they have even more than WT, which is left unexplained, 2) Casp1/11 dKO show similarly reduced levels of neutrophils and partially reduced levels of monocytes, and still do have high levels of CFUs. Maybe additional experiments that would more convincingly show a loss of a certain cell type due to pyroptosis could help. Or alternatively, such a loss could be introduced artificially (deplete neutrophils for example) and show that this is detrimental during infection.

Response: We thank the reviewer for this insight. We thought on the loss on the immune cell through pyroptosis but we do agree with the referee that is at this point speculative. Regarding NLRP3, ASC and Casp11 data, we believe that the data of these experiments needs to be interpreted in terms of balance between effector population recruitment and cell death. We also agreed with the reviewer that this balance recruitment/cell death partly explain the phenotype and cytokine overproduction is certainly the leading cause of death. Regarding the balance recruitment/cell death, for instance, the data in Fig3 a and b shows a similar level of neutrophils/monocytes recruitment in the lungs between NLRP3 and WT whereas in Fig4 a and b the cell death level is slightly lower or elevated leading to a slight CFU reduction in the lungs. By contrast Casp 1/11 dKO shows lower recruitment of effector cells in the lungs (Fig 3 a and b) whereas cell death is reduced (Fig 4 a and b) leading to a similar bacterial burden in the lungs (Suppl Fig 2c). Interestingly Caspase 11 single knockout shows similar or elevated recruitment of the effector cells (Fig 3 a and b) and reduced cell death (Figure 4 a and b) leading to a reduction of bacterial burden in the lungs (Figure 2c). We therefore believe it is a balance between recruitment and cell death. While we cannot definitely prove it and indeed it would require specific population depletion in future experiment by crossing with mouse knockout *ccr2* or *cd11* KO (the depletion experiments using antibodies did not work on the knockout unfortunately). To reflect on the referee's point, we added the following L255 "These data collectively suggest that in *Nlrp3^{-/-}*, *Asc^{-/-}*, *Casp11^{-/-}* and *Casp1/11^{-/-}* mice, the balance between effector recruitment (increase cell recruitment) and programmed cell death (lower cell death) might have contributed to lower bacterial burden and decreased host mortality" and L434 Our studies revealed that in addition to early recruitment to the target tissues, a balance between effector cell recruitment and their persistence is important to maintain the neutrophils/monocytes effector cells in the tissues. This balance might be partly responsible for modulating the host resistance against MDR *A. baumannii*. It therefore suggests the innate immune response against *A. baumannii* infection might be driven by a tight regulation between effector cell recruitment and programmed cell death mediated by inflammasome activation, controlled by NLRP3/ASC and caspase-11.

Role of GBPs:

a) The phenotype of the GBP1 KO mice mimics Casp11 deficiency, but the title of that chapter is confusing as it claims that mGBP1 protects the host – wouldn't it be better to say that it is detrimental?

Response. Apologies for the confusion. We changed the title as followed" Deficiency in IFN-inducible guanylate binding protein 1 (GBP1) protects the host against *A. baumannii*."

b) Also, I do not understand how the authors can conclude that mGBP1 does not protect via direct killing? I don't think that this is the correct conclusion from the data presented. The data show that changes in the LOS (or complete loss of LOS) do not reduce Casp11 activation. Since LPS is the only known ligand of both GBPs and caspase-11 (-4), the correct conclusions must be: 1) that mGBP1 does not recognize LOS, but some other signal stemming from A.b. infections, and 2) that Casp11 does not recognize LOS but some other signal stemming from A.b. infections. Such a conclusion is quite unexpected and would be highly controversial, and would need of course additional data to convince the field.

Response: We thank the referee for this interesting feedback. Previous works have demonstrated direct binding LPS-GBP1. In the context of mouse infection with *A.baumannii*, we haven't found such evidence. On the direct killing, the IC50 from our direct killing assay (Suppl Fig 8) using the full-length protein are over 40 µg/ml and therefore non physiological. It left us to conclude that direct killing is not the mechanism of protection. Other signal could be possible, but we haven't been able to provide empirical evidence for an additional ligand and as mentioned from the referee will require additional research. We were hesitant to add a sentence on it in our first version.

To reflect the points of this discussion, we 1/ removed the mention to direct killing in the manuscript (e.g L350) and 2/ added the following statement in the discussion section. "However we cannot eliminate that GBP1 and Caspase 11 do recognize another ligand (to be determined) from *A.baumannii* infection."

Minor points:

Fig 5d/e: The description of the data remain misleading,, as the authors do not point out that IFNAR deficiency has no impact on CFU in other organs save the lung.

Response: we altered the sentence by adding L289 "Interestingly, we observed a significant reduction in bacterial load in the lungs but not in the other organs (Fig. 5d-e)"

REVIEWERS' COMMENTS:

Reviewer #1 (Remarks to the Author):

The authors well addressed all issues raised by the reviewer.

Reviewer #2 (Remarks to the Author):

My concerns have been addressed.

Reviewer #3 (Remarks to the Author):

The authors present a much improved version of their manuscript, where they addressed the points raised by the referees. I have no additional comments.

Response: we would like to thank the referees for their comments and suggestions to improve the manuscript. We are grateful for their contributions in the peer-review process.